# Extracellular Pgk1 interacts neural membrane protein enolase-2 to improve the neurite outgrowth of motor neurons

Chuan-Yang Fu[1], Hong-Yu Chen[2], Cheng-Yung Lin[3], Shiang-Jiuun Chen[4,5], Jin-Chuan Sheu[6] & Huai-Jen Tsai [1,2,7 ✉]

Understanding the molecular interaction between ligand and receptor is important for providing the basis for the development of regenerative drugs. Although it has been reported that extracellular phosphoglycerate kinase 1 (Pgk1) can promote the neurite outgrowth of motoneurons, the Pgk1-interacting neural receptor remains unknown. Here we show that neural membranous Enolase-2 exhibits strong affinity with recombinant Pgk1-Flag, which is also evidently demonstrated by immunoelectron microscopy. The 325th-417th domain of Pgk1 interacts with the 405th-431st domain of Enolase-2, but neither Enolase-1 nor Enolase-3, promoting neurite outgrowth. Combining Pgk1 incubation and Enolase-2 overexpression, we demonstrate a highly significant enhancement of neurite outgrowth of motoneurons through a reduced p-P38-T180/p-Limk1-S323/p-Cofilin signaling. Collectively, extracellular Pgk1 interacts neural membrane receptor Enolase-2 to reduce the P38/Limk1/Cofilin signaling which results in promoting neurite outgrowth. The extracellular Pgk1-specific neural receptor found in this study should provide a material for screening potential small molecule drugs that promote motor nerve regeneration.

[1] Department of Life Science, Fu Jen Catholic University, New Taipei City, Taiwan. [2] Institute of Molecular and Cellular Biology, National Taiwan University, Taipei, Taiwan. [3] Institute of Biomedical Sciences, MacKay Medical College, New Taipei City, Taiwan. [4] Department of Life Science and Institute of Ecology and Evolutionary Biology, National Taiwan University, Taipei, Taiwan. [5] TechCommon-5, Bioimage Tool, College of Life Science, National Taiwan University, Taipei, Taiwan. [6] Liver Disease Prevention and Treatment Research Foundation, Taipei, Taiwan. [7] School of Medicine, College of Medicine, Fu Jen Catholic University, New Taipei, City, Taiwan. ✉email: 012102@mail.fju.edu.tw

Phosphoglycerate kinase 1(Pgk1) is an essential enzyme in the aerobic glycolysis pathway. Pgk1 catalyzes the reversible transfer of a phosphate group from 1,3-bisphosphoglycerate to ADP and produces 3-phosphoglycerate and ATP[1–3]. Pgk1 is one of the isoforms of the Pgk family, consisting of Pgk1 and Pgk2, both highly conserved throughout evolution in all organisms[4]. They have similar molecular structures and biological functions in human, encompassing 417 amino acids with 87–88% sequence identity[5]. However, Pgk1 and Pgk2 have different expression distribution. Pgk2, encoded by an autosomal gene, is only expressed during spermatogenesis, while Pgk1, located on the X-chromosome, is ubiquitously expressed in all cells[6]. It is clear that the protein structure of Pgk1 is monomeric, containing two similar-sized Rossmann fold domains, corresponding to the N- and C-terminal, respectively, connected by a hinge region through hydrophobic interactions and hydrogen bonds[2].

Besides canonical regulation of glycolytic metabolism, Pgk1 has other functions, including mediating autophagy initiation[7,8] and DNA replication and repair in mammalian cell nuclei[9]. However, it was recently discovered that low expression level of Pgk1 is related to many neurodegenerative diseases. For example, Parkinsonism patients exhibit Pgk1 deficiency which suggests that reduced glycolysis may contribute to nigrostriatal damage[10]. In contrast, enhanced Pgk1 activity, along with increasing glycolysis, slows down neurodegeneration and the symptoms of Parkinson's disease[11] and Amyotrophic lateral sclerosis[12]. It can also rescue motor axon phenotypes in spinal muscular atrophy zebrafish[13].

Although the intracellular function of Pgk1 is well-investigated[14,15], its extracellular function remains to be elucidated. Lin et al. reported that extracellular Pgk1 (ePgk1) has a noncanonical function independent of its glycolytic canonical role[16]. Specifically, it can be secreted from the muscle cells, and through decreased Rac1-GTP/p-Pak1-T423/p-P38-T180/p-MK2-T334/p-Limk1-S323 signaling, it then reduces p-Cofilin, which is a hallmark of growth cone collapse in neural cells[17,18]. These cascading events lead to enhanced neurite outgrowth of motor neurons. Interestingly, this ePgk1 secreted from skeletal muscle cells can enhance neurite outgrowth in motor neurons. Therefore, we speculate the presence of a membrane receptor specific to motor neurons that receives the signaling of secreted ePgk1 to then regulate the Pak1/P38/MK2/Limk1/Cofilin pathway axis, further affecting neurite outgrowth in motor neurons. Herein, we aimed to identify such motor neuron membrane receptor interacting with secretory ePgk1 protein, confirm the interaction between ePgk1 and such membrane receptor, and then determine if ePgk1 can, upon receptor binding, perform the signaling required for neurite outgrowth of motor neurons as described above.

In this study, we applied immunoprecipitation and LC-MS/MS analyses to screen putative membrane receptors of ePgk1. Out of 19 screened neural membrane proteins, we found three putative proteins potentially interacting with recombinant Pgk1-Flag. Among them, Enolase-2 (Eno2) had the strongest affinity with Pgk1-Flag, which was also supported by immunoelectron microscopy. Combining ePgk1 and overexpression of intracellular Eno2, but neither Eno1 nor Eno3, significantly promoted neurite outgrowth not only in neural NSC34 cells, but also in zebrafish embryos. Such interaction further decreased the downstream p-P38-T180/p-Limk1-S323/p-Cofilin axis of MAPK signal transduction in NSC34 cells. Collectively, these findings support the hypothesis that neural membrane receptor Eno2 receives the signaling of secreted Pgk1 and then regulates the P38/Limk1/Cofilin signaling axis, further improving the neurite outgrowth of motor neurons. Therefore, we propose that extracellular administration of Pgk1 could serve as a neuroprotective drug that might be potentially applied in degenerative diseases.

## Results

**Eno2, located at the neural membrane, possessed the highest binding affinity with secreted Pgk1, which was supported by direct observation on immunoelectron microscopy (IEM).** To search for the membrane receptor bound by secreted Pgk1, we first produced recombinant Pgk1 fused with Flag reporter (Pgk1-Flag) using the Baculovirus expression system. Recombinant Pgk1-Flag was then applied in the Flag pull-down assay for the membrane proteins extracted from NSC34 cells, followed by silver staining analysis (Fig. 1a). LC-MS/MS was then used to analyze the membrane proteins bound by Pgk1-Flag (Fig. 1a). Based on the 19 detected peptides and their scores presented on LC-MS/MS (Supplementary Table 1), three putative proteins, including Transient Receptor Potential Channel 5 (TRPC5), Toll-like Receptor 9 (Tlr9) and Eno2, potentially interacting with recombinant Pgk1-Flag, were selected for further study.

Next, we generated recombinant TRPC5, Tlr9 and Eno2 fused with Myc reporter using the Baculovirus expression system, followed by cell surface crosslinking-IP. The results demonstrated that Eno2, but not TRPC5 or Tlr9, interacted strongly with Pgk1 (Fig. 1b), suggesting that Eno2 might be the likely candidate protein bound by Pgk1.

Following up this experiment, we also employed cell surface crosslinking-IP to demonstrate the capability of Eno2 to interact with Pgk1 (Fig. 1c). Data based on the quantification of Pgk1 and Eno2 interaction revealed the results shown in Fig. 1d. First, when the intensity ratio of Myc level relative to Flag level for the Pgk1-Flag group was normalized as 1, that of the Pgk1-Flag plus Eno2-Myc group was 14-fold higher. Second, when the intensity ratio of Flag level relative to Myc level for the Eno2-Myc group was normalized as 1, that of the Eno2-Myc plus Pgk1-Flag group was eight-fold higher.

Besides the biochemical evidence shown above, we went further to prove whether ePgk1 could physically localize with endogenous Eno2 at the membrane of neural cells by direct observation on IEM. First, NSC34 cells were incubated in the medium containing recombinant EGFP-Flag, which served as a negative control. After performing double immunogold labeling on ultrathin sections of NSC cells, images showed that the immunostaining signal of Eno2 (12-nm gold particles) was presented on the cell membrane, while the immunostaining signal of EGFP-Flag (18-nm gold particles) presented randomly outside the Eno2 signal (Fig. 2a–d). No related evidence between these two signals was found. However, when NSC34 cells were incubated in medium containing recombinant Pgk1-Flag, the immunostaining signal of Eno2 (12-nm gold particles) and immunostaining signal of Pgk1-Flag (18-nm gold particles) were localized proximally on the cell membrane (Fig. 2e–h), suggesting that ePgk1 was tightly associated with Eno2 on the cell membrane of neural cells.

Based on direct observation through IEM, as shown in Fig. 2, we randomly selected nine signals from three signal-positive neural cells. We found that colocalization between Pgk1-Flag (18-nm gold particles) and Eno2 (12-nm gold particles) exhibited two patterns: (1) one large particle associated with one small particle (88.8 %, $n = 9$) and (2) one large particle associated with two small particles (11.2%, $n = 9$). The line of evidence suggested that one molecule of ePgk1 ligand could interact with either one monomer (major type) or one dimer (minor type) of receptor Eno2.

**The 405[th]–431[st] amino acid residues of membrane Eno2 interacted with ePgk1.** To determine whether a specific domain of membrane protein Eno2 of neural cells would interact with secreted Pgk1 from muscle cells, first, we used TMHMM-2.0 software (https://services.healthtech.dtu.dk/service.php?TMHMM-2.0)[19] to

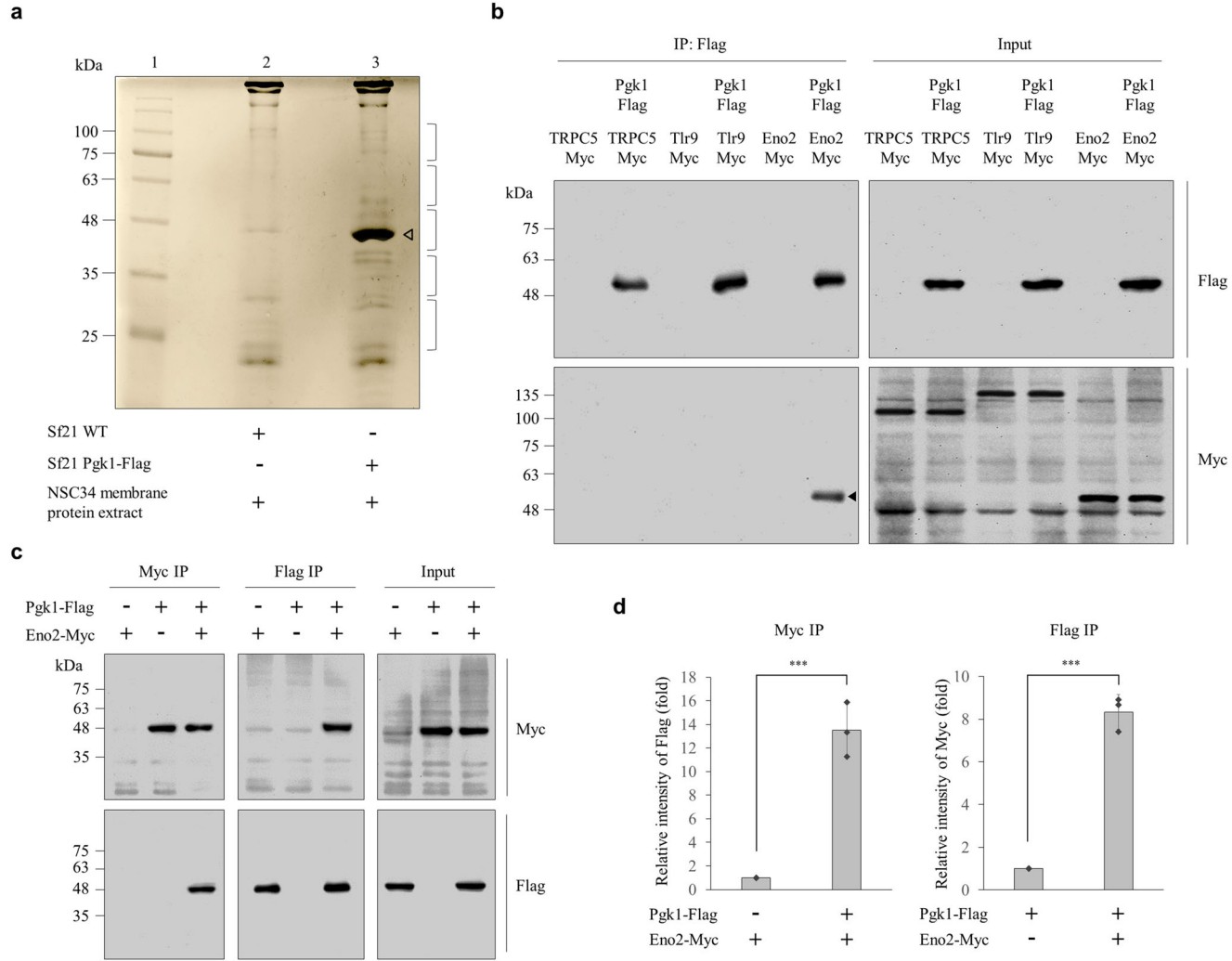

**Fig. 1 Screening and identification of putative membrane proteins that interact with mouse Pgk1. a** The pull-down assay. Membrane proteins extracted from NSC34 cells were pulled down by recombinant mouse Pgk1 fused with Flag (Pgk1-Flag) produced from Sf21 insect cells. The precipitated proteins were analyzed on SDS-PAGE by silver staining: lane 1, protein markers; lane 2, protein profiles of mixture containing Flag beads, extracts of Sf21 wild-type (WT) cells and membrane proteins extracted from NSC34 cells; and lane 3, protein profiles of mixture containing Flag beads, Pgk1-Flag produced by Sf21 and membrane proteins extracted from NSC34 cells. Location of the Pgk1-Flag band was marked with an empty arrow. Brackets on the right represented the gel slices excised for LC-MS/MS. **b** Three putative proteins that interacted with Pgk1-Flag were screened by crosslinking-immunoprecipitation (IP). The TRPC5-Myc, Tlr9-Myc and Eno2-Myc proteins were incubated in medium containing Pgk1-Flag, followed by cell surface crosslinking-IP using anti-Flag (IP: Flag) and then assessed by western immunoblot (IB) using either anti-Flag (Flag) or anti-Myc (Myc). Only transfection of Eno2-Myc was detected by a positive band, as marked with a solid arrow. Input represents 10% of the total cell extract used for each immunoprecipitation. **c** Cell surface crosslinking-IP to demonstrate the direct interaction between Pgk1-Flag and Eno2-Myc. After Eno2-Myc-expressing Sf21 cells were incubated in Pgk1-Flag-containing medium for 2 h, the cell lysate was immunoprecipitated with either anti-Flag (IP: Flag) or anti-Myc (IP: Myc), and western blot was performed using anti-Flag (Flag) or anti-Myc (Myc). Input represents 10% of the total cell extract used for each immunoprecipitation. Data were averaged from three independent experiments. **d** Quantification of the intensities of Flag and Myc shown on cell surface crosslinking-IP. The IP intensities of Pgk1-Flag and Eno2-Myc were individually normalized as 1. Data were averaged from three independent experiments and presented as mean ± SD ($n = 3$). Student's $t$ test was used to determine significant differences between each group (***$P < 0.001$). The same protein extracts from each experiment were loaded onto separate western blots and probed for individual proteins, in parallel.

analyze the transmembrane properties of Eno2, and showed that Eno2 displayed no transmembrane properties (Supplementary Fig. 1). Second, we used ProtScale-ExPASy software (https://web.expasy.org/protscale/)[20] to analyze the hydropathicity of Eno2 and found the 405th–434th amino acid region located at the C-terminus of Eno2 to be highly hydrophilic (Supplementary Fig. 2). It has been reported that the C-terminus of Eno2 promotes neuronal survival, differentiation and axonal regeneration[21]. Eno1, another membrane protein in the Enolase family, has been reported to interact with plasminogen through its exposure to C-terminal lysine residue[22,23]. When we also used ProtScale-ExPASy software to analyze its

relative hydrophilicity, we found its C-terminal domain structure to be hydrophilic, similar to that of Eno2 (Supplementary Fig. 2a, b). Extrapolating from this evidence, it was hypothesized that the hydrophilic C-terminus 405th–434th of Eno2, a peripheral protein located on the cell membrane, might also be a domain that interacts with extracellular ligand Pgk1 we studied here.

To test this hypothesis, we engineered Eno2 with C-terminal deletion and fused with Myc, including (a) Eno2(Δ405–431)-Myc, in which the 405–431 domain at C-terminus of Eno2 was deleted, except a PDZ-binding motif, and (b) Eno2(Δ432–434)-Myc, in which a PDZ-binding motif of Eno2 was deleted. The Baculovirus

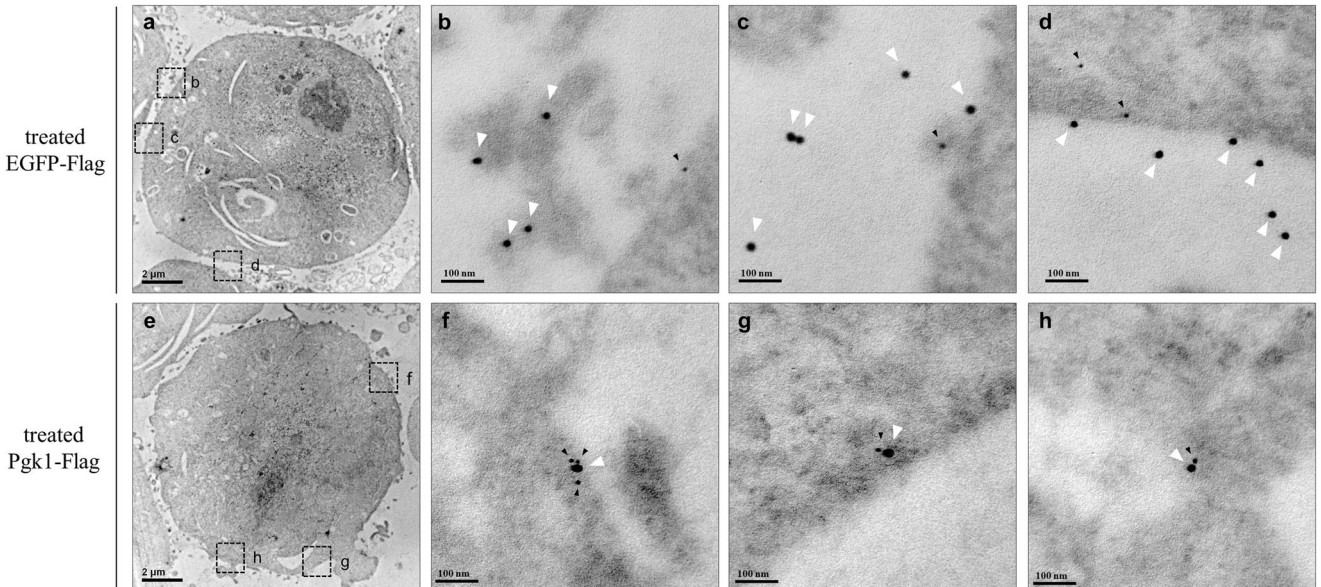

**Fig. 2 IEM observation revealed that extracellular Pgk1 and endogenous membrane protein Eno2 were located closely on the cell surface of neural cells. a–h** NSC34 neural cells were incubated in the medium containing either EGFP-Flag (**a–d**), which served as a negative control, or recombinant Pgk1-Flag (**e–h**). Double immunogold labeling was performed on the ultrathin sections of cells and examined at low magnification (X 8000) (**a, e**). **b–d, f–h** were amplified at high magnification (X 80,000) from the rectangular areas indicated within (**a, e**), respectively. Ultrathin sections were reacted simultaneously with mouse anti-Eno2 antibody and rabbit anti-Flag antibody, followed by 12-nm gold-conjugated anti-mouse IgG antibody (black arrowheads) and 18-nm gold-conjugated anti-rabbit IgG antibody (white arrowheads). Scale bar indicates 2 μm (**a, e**) and 100 nm (**b–d, f–h**). Data were averaged from three independent experiments.

system was used to generate the above two recombinant proteins (Fig. 3a), and it was found that the Pgk1-Flag signal could be pulled down by Eno2-Myc and Eno2(Δ432–434)-Myc, but not Eno2(Δ405–431)-Myc (Fig. 3b). After quantifying the interaction between Pgk1 and both truncated recombinant proteins, we found that the 405th–431st amino acid residues of Eno2 comprise the key domain that interacts with ePgk1 (Fig. 3c), thus confirming our hypothesis.

**The 325th–417th amino acid residues of ePgk1 interacted with membrane Eno2.** To elucidate the domain of ePgk1 that interacts with membrane Eno2, we performed Pgk1 domain mapping and generated various truncated domains of Pgk1 fused with Flag, including fragments 1–145, 146–417, 225–417, and 325–417, using the Baculovirus expression system (Fig. 4a). Domain mapping analysis revealed that the Eno2-Myc signal could be pulled down by all truncated forms of Pgk1-Flag, except the 1–145 fragment (Fig. 4b). Thus, we ruled out the possibility that the 1–145 domain could interact with Eno2. Furthermore, when the relative binding intensity between Pgk1-Flag and Eno2-Myc was quantified, we found that the relative binding intensities from fragments 1–417, 146–417, 225–417, and 325–417 of Pgk1 were not significantly different, as long as the intensity ratio of Myc level relative to Flag level for the Sf21 WT plus Eno2-Myc was normalized as 1 (Fig. 4c). Since the domain including 325th–417th amino acid residues (325–417 domain) of ePgk1 were shared by all examined truncated forms displaying positive reaction, we concluded that the 325–417 domain of ePgk1 was the key domain interacting with Eno2.

**Combining Pgk1 incubation and Eno2 overexpression promoted neurite outgrowth of motor neurons derived from neural cells through the reduction of p-Cofilin.** Eno2 is one member of the Eno superfamily which includes three isoenzymes, Eno1, 2, and 3, sharing high homology of amino acid sequence[21].

We investigated the interaction between each member of the Eno superfamily and Pgk1 relative to reduced p-Cofilin and subsequent enhancement of neurite outgrowth of motor neurons. To accomplish this, we first employed an in vitro system in which we used NSC34 neural cells with two experimental groups and one untreated control group. One experimental group consisted of cells incubated with recombinant Pgk1-Flag alone, while another group consisted of cells incubated with recombinant Pgk1-Flag combined with transfection mRNA encoding Eno1, Eno2, or Eno3. Compared to untreated control neural cells, we found that the proportion of neurite-bearing cells and average neurite length of neural cells incubated with Pgk1 alone displayed an increase of 7% and 12.14 μm, respectively (Fig. 5a–d). Moreover, the proportion of neurite-bearing cells and average neurite length of neural cells incubated with Pgk1 together with Eno2 transfection displayed an increase of 15% and 40.34 μm, respectively (Fig. 5a–d). Interestingly, compared to cells incubated with Pgk1 alone, neurite-bearing cells and average neurite length were not significantly increased in cells incubated with Pgk1 together with transfection of either Eno1 or Eno3 (Fig. 5a–d).

We also extracted total proteins from NSC34 cells incubated with Pgk1 and simultaneously transfected with *Eno1, 2,* or *3* mRNA, followed by western blot analysis. Results demonstrated that the protein level of p-Cofilin in NSC34 cells incubated with Pgk1 was significantly decreased compared with control cells only transfected with pCS2 (Fig. 5e, f), as we expected. However, the protein level of p-Cofilin was significantly downregulated when NSC34 cells were treated with both Pgk1 incubation and *Eno2* mRNA transfection, but not when Pgk1 was incubated with either *Eno1* or *Eno3* mRNA transfection (Fig. 5e, f).

Furthermore, we designed siRNAs to specifically knock down Eno1 and Eno2 to further confirm the hypothesis concluded above. It has been reported that Eno3 expression is muscle-specific[24,25]; therefore, we herein only focused on Eno1 and Eno2. First, we examined whether the expression of Eno1 and Eno2 could be reduced in NSC34 cells transfected with *Eno1*-siRNA

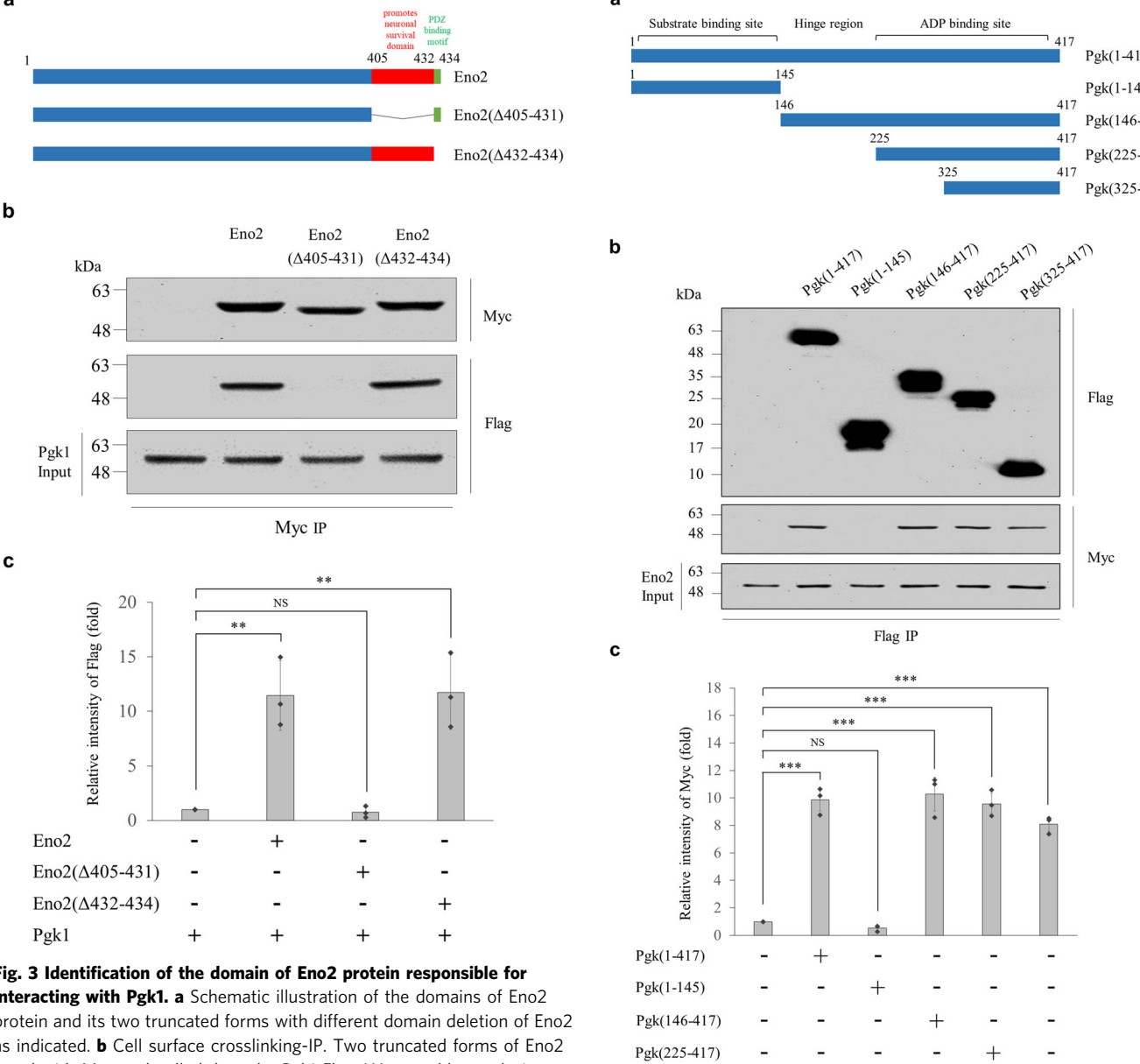

**Fig. 3 Identification of the domain of Eno2 protein responsible for interacting with Pgk1. a** Schematic illustration of the domains of Eno2 protein and its two truncated forms with different domain deletion of Eno2 as indicated. **b** Cell surface crosslinking-IP. Two truncated forms of Eno2 fused with Myc and pulled down by Pgk1-Flag. Western blot analysis was used to detect the presence of Pgk1-Flag and Eno2-Myc by anti-Flag and anti-Myc antibody, respectively. Data were averaged from three independent experiments. **c** Quantification of immunoprecipitation intensity shown on cell surface crosslinking-IP. The immunoprecipitation intensity of Eno2-Myc was quantified based on the intensity of Pgk1-Flag normalized as 1. Data were averaged from three independent experiments and presented as mean ± SD (n = 3). One-way ANOVA, followed by Tukey's multiple comparison test, was used to perform statistical analysis (**P < 0.005 and NS, not significant, P > 0.05). The same protein extracts from each experiment were loaded onto separate western blots and probed for individual proteins, in parallel.

and *Eno2*-siRNA, respectively. Compared to control siRNA, western blot analysis demonstrated that the protein levels of Eno1 and Eno2 extracted from NSC34 cells transfected with *Eno1*- and *Eno2*-siRNAs were reduced by 47% and 81%, respectively (Supplementary Fig. 3a, b), suggesting that *Eno1*- and *Eno2*-siRNAs used in this study could effectively block Eno1 and Eno2 expression, respectively. Second, we analyzed the total proteins extracted from NSC34 (a) transfected with control siRNA, (b) incubated with Pgk1, (c) transfected with *Eno1*-siRNA and

**Fig. 4 The domains of Pgk1 interacting with Eno2. a** Schematic illustration of full-length mouse Pgk1 (Pgk) and its various truncated fragments with different deletions. **b** Cell surface crosslinking-immunoprecipitation (IP) followed by western blot analysis. Various truncated forms of Pgk with Flag-tag, as indicated, were pulled down by Eno2-Myc. The fusion proteins of Pgk-Flag and Eno2-Myc were detected by immunoblotting (IB) using anti-Flag and anti-Myc antibody, respectively. The truncated forms that included the 325–417 segment of mouse Pgk1 [Pgk (325–417)] were those having the conserved domain that directly interacts with Eno2. Data were averaged from three independent experiments. **c** Quantification of cell surface crosslinking-IP. IP intensities of Pgk1-Flag were quantified based on the intensity of Eno2-Myc normalized as 1. Data were averaged from three independent experiments and presented as mean ± SD (n = 3). One-way ANOVA, followed by Tukey's multiple comparison test, was used to perform statistical analysis (***P < 0.001 and NS, not significant, P > 0.05). The same protein extracts from each experiment were loaded onto separate western blots and probed for individual proteins, in parallel.

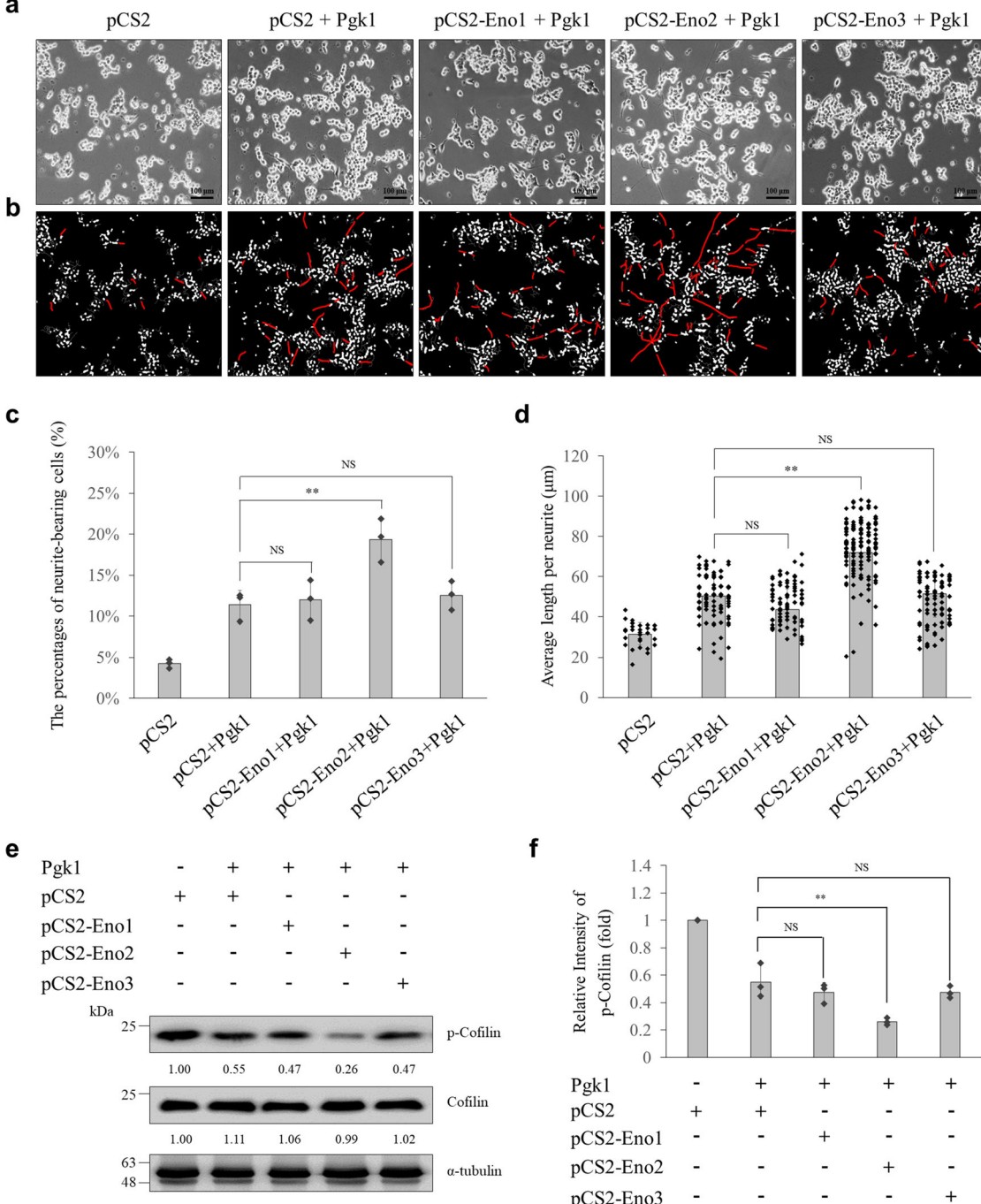

**Fig. 5 Extracellular Pgk1 interacting with membrane protein Eno2, but neither Eno1 nor Eno3, caused the reduction of p-Cofilin, resulting in enhancing neurite outgrowth of motoneurons derived from neural cells. a** Morphology of neurites derived from NSC34 neural cells was observed under microscopy. Five experimental groups were categorized: (1) cells treated with pCS2 transfection, (2) pCS2 transfection plus Pgk1 incubation (pCS2+Pgk1), (3) pCS2-Eno1 transfection plus Pgk1 incubation (pCS2-Eno1+Pgk1), (4) pCS2-Eno2 transfection plus Pgk1 incubation (pCS2-Eno2+Pgk1), and (5) pCS2-Eno3 transfection plus Pgk1 incubation (pCS2-Eno3+Pgk1). Scale bar, 100 μm. **b** *Neurite Outgrowth Analysis Application* Module of MetaMorph software was used to count neurite-bearing cells and measure neurite lengths. Cell body was marked in white, while neurite was marked in red. **c, d** Statistical analysis: **c** percentage of neurite-bearing cells among examined cells and **d** average neurite length of each group, as indicated. Data were averaged from three independent experiments and presented as mean ± SD ($n = 3$). One-way ANOVA, followed by Tukey's multiple comparison test, was used to perform statistical analysis (**$P < 0.005$ and NS, not significant, $P > 0.05$). **e** Western blot analysis. NSC34 cells incubated either in the presence (+) or absence (−) of Pgk1 and transfected either pCS2-Eno1-, -Eno2-, or -Eno3-Myc, followed by analyzing the levels of phosphorylated Cofilin at S3 (p-Cofilin) and total Cofilin. The α-tubulin served as an internal control. The change (in fold) of relative intensity of each examined protein against α-tubulin of each group compared to that of the pCS2-control group relative to each internal control set as 1 was presented under each lane. Data were averaged from three independent experiments. **f** Statistical analysis. Data were averaged from three independent experiments and presented as mean ± SD ($n = 3$). One-way ANOVA, followed by Tukey's multiple comparison test, was used to perform statistical analysis (**$P < 0.005$ and NS, not significant, $P > 0.05$). The same protein extracts from each experiment were loaded onto separate western blots and probed for individual proteins, in parallel.

incubated with Pgk1 and (d) transfected with *Eno2*-siRNA and incubated with Pgk1. Compared with the cells transfected with control siRNA, the protein level of p-Cofilin was decreased in the cells incubated with Pgk1 (Supplementary Fig. 3c, d), supporting our previous conclusion that the addition of Pgk1 could improve neurite outgrowth. Nevertheless, compared with the cells incubated with Pgk1, the p-Cofilin level remained unchanged in cells transfected with *Eno1*-siRNA and incubated with Pgk1, while the p-Cofilin level was increased in the cells transfected with *Eno2*-siRNA and incubated with Pgk1 (Supplementary Fig. 3c, d), suggesting that the reduction of p-Cofilin induced by Pgk1 resulted from specific combination between ePgk1 and Eno2, but not Eno1.

We also employed cell surface crosslinking-IP to determine whether ePgk1 could interact with Eno1. After quantifying the intensities of interactions between ePgk1 and Eno1 proteins and between ePgk1 and Eno2 proteins, we found that ePgk1 could only interact with Eno2, but not Eno1 (Supplementary Fig. 4a, b), thus confirming our hypothesis.

**Either overexpression or attenuation of ePgk1 and eno2 affects the phenotype of trunk motor neurons in zebrafish embryos.** We further performed an in vivo system using zebrafish transgenic line *Tg(mnx1:GFP)* in which motor neurons were tagged by GFP[26]. We designed two experimental groups and one untreated control group. In one experimental group, embryos were incubated with recombinant zebrafish Pgk1(zPgk1)-Flag alone. In the second experimental group, embryos were incubated with zPgk1-Flag combined with injection of *eno1, eno2* or *eno3* mRNA. Compared to untreated control embryos at 30 hpf (Fig. 6a–c), we found that 48% ($n = 45$) of embryos incubated with zPgk1-Flag alone displayed ectopically branched neurons, while 87% ($n = 43$) of embryos incubated with zPgk1-Flag together with *eno2* mRNA injection displayed ectopically branched neurons (Fig. 6a–c). Nevertheless, compared to embryos incubated with zPgk1-Flag recombinant protein alone, the percentage of ectopically branched neurons was not increased in embryos incubated with zPgk1-Flag together with either *eno1* or *eno3* mRNA, which was 47% ($n = 43$) and 43% ($n = 45$), respectively (Fig. 6a–c).

Furthermore, we examined the protein level of p-Cofilin in zebrafish embryos from transgenic line *Tg(mnx1:GFP)* by western blot analysis. Results demonstrated that the protein level of p-Cofilin in zebrafish embryos incubated with zPgk1-Flag was decreased compared with untreated control embryos (Fig. 6d, e), while that of zebrafish embryos was significantly downregulated in embryos treated with both zPgk1-Flag incubation and *eno2* mRNA injection, but not zPgk1-Flag incubation with either *eno1* or *eno3* mRNA injection (Fig. 6d, e).

Next, we used a loss-of-function strategy to prove that neurite outgrowth of motor neurons is impacted by the interaction between extracellular zPgk1-Flag and eno2 membrane protein. We injected *eno2*-MO into one-celled zebrafish embryos to knock down eno2 expression. In comparison to control MO-injected zebrafish embryos which was normalized as 1, western blot analysis demonstrated that the protein level of eno2 was reduced by 80% and the protein level of p-Cofilin was significantly increased to 254% in the *eno2*-MO-injected zebrafish embryos (Supplementary Fig. 5a), suggesting that *eno2*-MO is specific and effective in knocking down the expression of eno2 further to resulting increase of p-Cofilin.

Furthermore, compared to two control groups, i.e., uninjected embryos and embryos injected with control MO, 51% ($n = 45$) of embryos incubated with zPgk1 protein alone exhibited ectopic outgrowth of motor neurons (Supplementary Fig. 5b, c), consistent with the result shown in the above section. The percentages of

embryos exhibiting ectopic outgrowth of motor neurons were not significantly different between embryos incubated with zPgk1 plus injected with control MO and embryos incubated with zPgk1 protein only (Supplementary Fig. 5b, c). However, the percentage of ectopic phenotype of motor neurons shown in the embryos incubated with ePgk1 and injected with *eno2*-MO was greatly reduced to 29% (Supplementary Fig. 5b, c). This line of evidence suggested that the specific interaction between ePgk1 and membrane protein Eno2 does reduce p-Cofilin, resulting in increased neurite outgrowth of motor neurons in zebrafish embryos.

**The signaling pathway involved in reducing p-Cofilin through the interaction between ePgk1 and membrane protein Eno2 in NSC34 cells.** We asked whether the signaling pathway involved in the enhancement of neurite outgrowth and the decrease of p-Cofilin, as mediated by the interaction between ePgk1 and membrane Eno2, is identical to that reported in ref. [16], in which they demonstrated that secreted Pgk1 reduces p-Cofilin expressed in NSC34 cells through the p-P38-T180/p-Limk1-S323 signals of the MAPK transduction pathway. Therefore, to address this question, we extracted the total proteins of NSC34 cells cultured with Pgk1 and/or transfected with *Eno2* mRNA, followed by western blot analysis. Compared with control cells only transfected with pCS2, results showed that the protein level of p-Cofilin in NSC34 cells either incubated with Pgk1 or transfected with Eno2 was significantly decreased (Fig. 7a, b). Interestingly, we found that the protein level of p-Cofilin was significantly downregulated in NSC34 cells cultured in medium with both Pgk1 addition and *Eno2* mRNA transfection (Fig. 7a, b), suggesting that the interaction between ePgk1 and membrane Eno2 resulted in a synergism that affected the downmodulated expression of p-Cofilin.

We then continued to determine the signaling pathway involved in reducing p-Cofilin through the interaction between ePgk1 and membrane Eno2 in NSC34 cells. We found that (a) the protein levels of p-P38-T180 and p-Limk1-S323 were decreased in total protein lysate of NSC34 cells, either incubated with ePgk1 or transfected with *eno2* mRNA; (b) the protein levels of p-P38-T180 and p-Limk1-S323 were synergistically decreased in the lysate of NSC34 cells treated with the combination of Pgk1 incubation and *eno2* mRNA transfection; and (c) the protein level of p-Limk1-T508 remained unchanged in the lysate of NSC34 cells treated with the combination of Pgk1 incubation and *eno2* mRNA transfection.

Next, to determine if the above signaling pathway that regulates neural differentiation of NSC34 cells was affected by adding an Eno2-specific antibody to block Eno2, we employed cell surface crosslinking-IP. According to the results, the binding between Pgk1-Flag and Eno2-Myc was not detected by cell surface crosslinking-IP in the presence of Eno2 antibody (Supplementary Fig. 6). This finding suggests that the interaction between ligand Pgk1 and receptor Eno2 would fail if Eno2 was blocked by Eno2-specific antibody.

We then collected the total proteins of NSC34 cells extracted from three sources, including (a) cells incubated with normal mouse IgG, which served as a control group; (b) cells incubated with Pgk1 and IgG; and (c) cells incubated with Pgk1 plus Eno2 antibody. Compared with control cells incubated with normal mouse IgG normalized as 1, the protein levels of p-P38-T180, p-Limk1-S323, and p-Cofilin were all significantly reduced in cells incubated with Pgk1 and IgG by 61%, 30%, and 43%, respectively (Fig. 7c, d). However, the levels of total proteins of P38, p-Limk1-T508, Limk1 and Cofilin remained relatively unchanged in the Pgk1 plus IgG cells (Fig. 7c, d), suggesting that the effect of ePgk1 on reducing p-Cofilin was functional in the presence of general antibody. Nevertheless, compared to the IgG control group, the

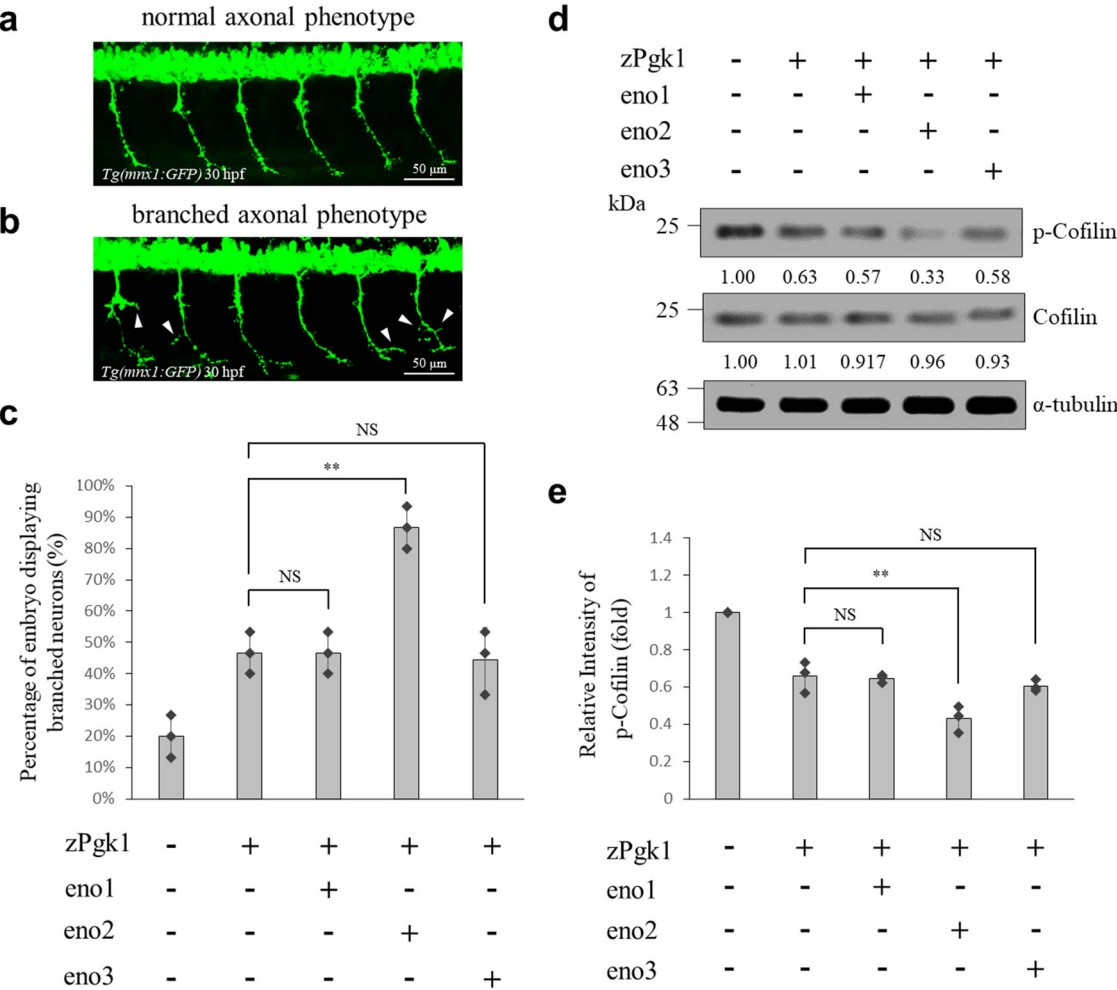

**Fig. 6 Extracellular Pgk1 interacts with eno2 on the neural membrane to reduce p-Cofilin, enhancing neurite outgrowth of motor neurons in zebrafish embryos. a, b** Phenotypes of motor neurons observed in the embryos from transgenic line *Tg(mnx1:GFP)*: **a** normal axonal phenotype neurons; **b** branched axonal phenotype neurons (indicated by white arrows). Scale bar: 50 μm. **c** After embryos were incubated, either with (+) or without (−) zebrafish Pgk1 for 24 h, they were injected with or without three isoforms of *eno*-mRNAs. The percentage of embryos displaying branched motor neuron phenotype among the total number (*n*) of examined embryos in each group was calculated. Data were averaged from three independent experiments. Data were averaged from three independent experiments and presented as mean ± SD (*n* = 3). One-way ANOVA, followed by Tukey's multiple comparison test, was used to perform statistical analysis (**$P < 0.005$ and NS, not significant, $P > 0.05$). **d** Western blot analysis. Zebrafish embryos from transgenic line *Tg(mnx1:GFP)* were incubated with (+) or without (−) zebrafish Pgk1 for 24 h and injected with or without three isoforms of *eno*-mRNAs. Non-injected embryos served as control, followed by collecting total protein lysate from *Tg(mnx1:GFP)* and then analyzing the levels of phosphorylated Cofilin at S3 (p-Cofilin) and total Cofilin. The α-tubulin served as an internal control. The change (in fold) of relative intensity of each examined protein against internal marker α-tubulin compared to that of the zPgk1-control group relative to each internal control set as 1 was presented below each lane. Data were averaged from three independent experiments. **e** Statistical analysis. The change (in fold) of relative intensity of p-Cofilin against internal marker tubulin compared to that of the control group which was set as 1. Data were averaged from three independent experiments and presented as mean ± SD (*n* = 3). One-way ANOVA, followed by Tukey's multiple comparison test, was used to perform statistical analysis (**$P < 0.005$ and NS, not significant, $P > 0.05$). The same protein extracts from each experiment were loaded onto separate western blots and probed for individual proteins, in parallel.

levels of the above phosphorylated and total proteins remained nearly unchanged in cells incubated with Pgk1 plus Eno2 antibody, except that the levels of p-P38-T180 and p-Cofilin were reduced by 11% and 10%, respectively (Fig. 7c, d). Next, we extracted total proteins from (d) NSC34 cells incubated with Eno2 antibody alone, and the results showed that the protein levels of p-P38-T180, p-Limk1-S323, and p-Cofilin were all significantly increased (Fig. 7c, d) by 113%, 77%, and 62%, respectively, compared to those of control IgG-incubated cells. Collectively, these results showed that the reduction of p-P38/p-Limk1-S323/p-Cofilin axis by ePgk1 was mostly abolished in the presence of Eno2 antibody, further indicating that receptor Eno2 on neural membrane is specifically bound by ligand ePgk1 to trigger the signaling pathway that causes neurite outgrowth.

To further confirm this hypothesis, we employed *Eno2*-siRNA to specifically knock down endogenous Eno2, compared with cells transfected with control siRNA, the protein levels of p-P38-T180, p-Limk1-S323, and p-Cofilin were all significantly reduced by 58%, 34%, and 45%, respectively, while the total protein levels of p38, Limk1, and Cofilin, as well as p-Limk1-T508, remained nearly unchanged in cells incubated with Pgk1 and control siRNA (Supplementary Fig. 7a, b), suggesting that the reduction of p-P38/p-Limk1-S323/p-Cofilin axis by ePgk1 was functional in the presence of control siRNA. Moreover, we found that the levels of the above phosphorylated could be restored and total proteins remained unchanged in cells incubated with Pgk1 plus *Eno2*-siRNA except that p-Cofilin was reduced by 7% and p-P38 was increased by 14% (Supplementary Fig. 7a, b), suggesting that the

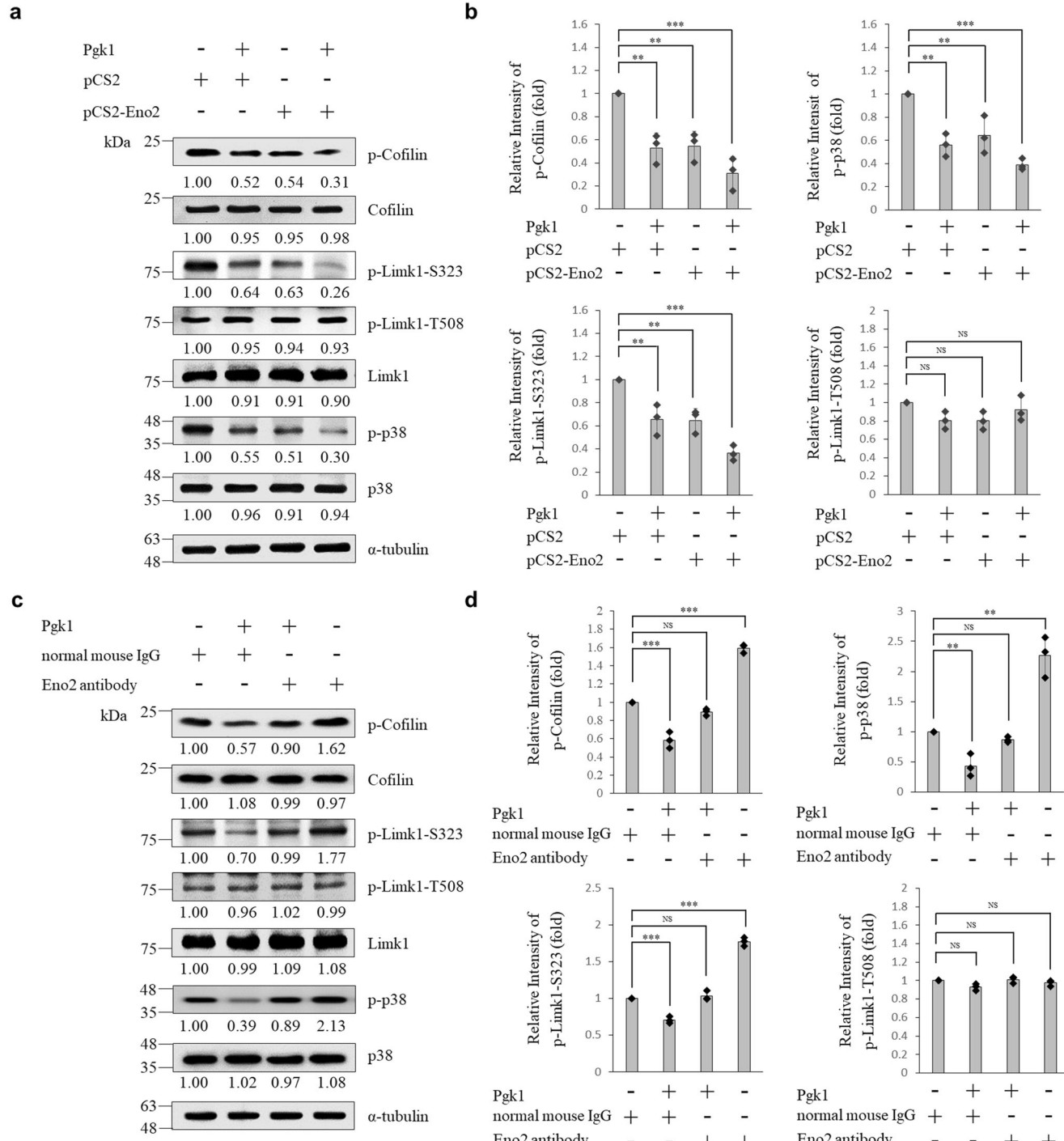

**Fig. 7 Expression levels of regulatory proteins controlling the production of phosphorylated Cofilin at S3 within NSC34 cells. a** Western blot analysis. NSC34 cells were incubated in the presence (+) or absence (−) of mouse Pgk1 and pCS2-Eno2-Myc transfection, followed by analyzing the levels of regulatory proteins, as indicated, involved in the signaling pathway that controls the production of phosphorylated Cofilin at S3. The α-tubulin served as an internal control. The change (in fold) of relative intensity of each examined protein against α-tubulin of each group compared to that of the pCS2-control group relative to each internal control set as 1 was presented at each lane. Data were averaged from three independent experiments. **b** Quantitative and statistical analyses. Data were averaged from three independent experiments and presented as mean ± SD ($n = 3$). One-way ANOVA, followed by Tukey's multiple comparison test, was used to perform statistical analysis (**$P < 0.005$, ***$P < 0.001$ and NS, not significant, $P > 0.05$). **c** Western blot analysis. NSC34 cells were incubated in the presence (+) or absence (−) of Pgk1, normal mouse IgG and Eno2 antibody, followed by analyzing the levels of regulatory proteins, as indicated. The α-tubulin served as an internal control. The change (in fold) of relative intensity of each examined protein against α-tubulin of each group compared to that of the normal mouse IgG-control group relative to each internal control set as 1 was presented at each lane. Data were averaged from three independent experiments. **d** Quantitative and statistical analyses. Data are calculated from three independent experiments and presented as mean ± SD ($n = 3$). One-way ANOVA, followed by Tukey's multiple comparison test was used to perform statistical analysis (**$P < 0.005$, ***$P < 0.001$ and NS, not significant, $P > 0.05$). The same protein extracts from each experiment were loaded onto separate western blots and probed for individual proteins, in parallel.

reduction of p-Cofilin by ePgk1 was nearly abolished by the presence of *Eno2*-siRNA. These results were consistent with those obtained from cells treated with Eno2 antibody, as described above. Meanwhile, when we extracted total proteins from NSC34 cells incubated with *Eno2*-siRNA alone, we found that the protein levels of p-P38-T180, p-Limk1-S323, and p-Cofilin were all significantly increased by 84%, 59%, and 54%, respectively, compared to those of control siRNA-transfected cells.

## Discussion

The well-known canonical function of the Enolase family is involved in glycolysis, serving as an energy supplier. These glycolytic enzymes are expressed in cytoplasm and consist of three isozymes: Eno1 is expressed ubiquitously, Eno2 expression is neuron-specific, and Eno3 expression is muscle-specific[21,22]. Also reported is a noncanonical function of Enolases associated with the localization of plasma membrane. For example, Plasminogen secreted from the liver cells can specifically bind to Eno1 on rat neuronal plasma membrane[27]. Meanwhile, Eno2 could be translocated toward the synaptic plasma membrane[28] and found in the synaptic plasma membrane of rat brain[29]. Based on IEM images shown in this study, we demonstrated that Eno2 is localized on the plasma membrane of neural cells, consistent with the evidence just cited. Using double-immunolabeling on IEM images, we further demonstrated that membranous Eno2, when labeled by 12-nm colloidal AuNPs, was physically associated with extracellular recombinant Pgk1-Flag labeled by 18-nm colloidal AuNPs. This provides direct evidence that membranous Eno2 serves as a receptor that interacts with ePgk1 to enhance the neurite outgrowth of motor neurons, as supported by our CO-IP results. Nevertheless, it has been reported that Eno2 is a dimeric protein composed of two isozymes: γγ and αγ[30,31]. The γγ isoform is found predominantly in cytoplasm of mature neurons, while the αγ isozyme is found in non-neuronal cells, such as glial cells[32]. Unexpectedly, based on direct observation through the double-immunolabeling shown on IEM, we notice that the major pattern (around 90% of examined signal-positive cells) exhibited the interaction between Pgk1-Flag and Eno2 was one molecule of Pgk1-Flag associated with one monomer of Eno2. Therefore, the topographical structure of interaction between ligand ePgk1 and receptor Eno2 is still controversial which remains to be clearly elucidated in the future.

Further addressing its protein structure, the noncanonical domain of membranous Eno1 has been reported to bind plasminogen through its exposed C-terminal lysine residue[22,23]. Using a primary culture of embryonic rat neocortical neurons, Hattori et al.[21] reported that the synthetic 30-amino-acid peptide corresponding to the 404th–433rd amino acid residues (404–433 domain) at C-terminus of Eno2 promotes neuronal survival. As further reported in ref. [33], when this 404–433 domain was incubated with neuroblastoma cells, neuronal survival, proliferation and differentiation were significantly improved. However, these results were not achieved when the same experiment was performed under two conditions causing loss of cellular internalization: (1) 404–433 domain without containing 432–433 or (2) 404–433-domain-coated beads. Nevertheless, in the present study, when we employed cell surface crosslinking-immunoprecipitation and domain mapping analysis of Eno2, the Pgk1-Flag signal could be pulled down by Eno2-Myc and Eno2(Δ432–434)-Myc, but not by Eno2(Δ405–431)-Myc. These results suggest that the 405–431 domain of Eno2 comprise the key motif contributing to improved neurite outgrowth and that such improvement is independent of the PDZ-binding motif (PBM, 432–434: S–V–L), which, upon its interaction with γ-1-syntrophin, transports Eno2 toward the plasma membrane[33]. As noted above, since the interaction between

extracellular Pgk1-Flag and membranous Eno2(Δ432–434)-Myc could trigger downstream signaling pathways to promote neurite outgrowth, we conclude if the truncated 405–431 domain of Eno2 is the key domain of Eno2, which serves as an interacting receptor of ePgk1 at the plasma membrane, then that could otherwise induce neurite outgrowth is not included as an internalization signal (Valylleucine (V–L) dipeptide), as proposed by Hafner et al.[34]. Taken together, the 434-amino-acid intact Eno2 protein displays at least three segments with their distinct characteristics: (1) the 405th–431st segment, a key biologically active domain, allows intact membranous Eno2 to perform Eno2/ePgk1 interaction which results in neurite outgrowth, as proposed by this study; (2) the 433rd–434th segment, an internalization signal (V–L), allows the synthetic 30-amino acid peptide corresponding to 404–434 domain of Eno2 to internalize into host cells, proposed in ref. [34]; and (3) the 432nd–434th segment, a PBM trafficking signal (S–V–L), allows the transport of Eno2 to plasma membrane through interaction with γ-1-syntrophin, as also proposed in ref. [33].

According to Bai et al.[35], who employed zebrafish transgene line *Tg(eno2:GFP)Pt404*, GFP signals revealed segmental spinal nerve roots, displaying either ventrally and dorsally or extending rostrally and caudally from spinal cord to innervate somatic muscles and teguments. They also reported that *eno2* mRNA expressed at 24 hpf and continuously presented in motor neurons until adulthood. Therefore, we designed an experiment to observe the axonal development of CaP motor neurons and counted the number of branched neurites in embryos at 30 hpf under fluorescence microscopy. Using zebrafish embryos as an in vivo system, we also demonstrated that the number of branched axonal neurites was significantly increased up to 87% of embryos after injection with *eno2* mRNA and incubation with ePgk1 protein.

Lin et al.[16] demonstrated that the addition of ePgk1 triggers a reduction in p-Cofilin, a hallmark of growth cone collapse, through downregulating the Rac-GTP/p-Pak1-T423/p-P38-T180/ p-MK2-T334/p-Limk1-S323 axis, in turn enhancing the neurite outgrowth of motor neurons. Here, we found that this signal transduction pathway is also involved in the interaction between membranous Eno2 receptor and ePgk1 ligand (Eno2/ePgk1 interaction) to control neurite outgrowth of motor neurons. Specifically, Eno2/ePgk1 interaction downregulates phosphorylation of P38 MAPK signaling via the P38/Limk1/Cofilin axis. In fact, many studies have found that P38 MAPK signaling is an important effector of motor neuron regulation. For example, the abnormal activation of P38 MAPK in SOD1-mutated mice induces motor neuron death[36]. In contrast, inhibition of P38 MAPK can rescue axonal retrograde transport defects, resulting in the promotion of cell survival[37]. In addition, the MKK6/P38/ MK2 pathway has been found to phosphorylate Limk1 at S323 to promote the phosphorylation of Cofilin in endothelial cells[38]. Furthermore, Lin et al.[16] demonstrated that secreted Pgk1 reduces p-Cofilin expressed in NSC34 cells through decreased p-P38/ p-Limk1-S323 signaling. This line of evidence supports our conclusion that Eno2/ePgk1 interaction controls neurite outgrowth of motor neurons by triggering the reduction of the p-P38/p-Limk1-S323/p-Cofilin axis.

Moreover, the positive effect of ePgk1 on the neurite outgrowth of motor neurons is not mediated through decreasing the p-ROCK2-Y256/p-LimK-T508/p-Cofilin pathway, as proposed in ref. [39] and ref. [40]. They reported that the Nogo66 domain of oligodendrocyte inhibits neuronal growth through Nogo66/NgR interaction. In this study, we revealed that enhanced neurite outgrowth of motor neurons, as mediated by Eno2/ePgk1 interaction, results from downregulating the p-P38-T180/p-Limk1-S323/p-Cofilin transduction pathway, not the p-ROCK2-Y256/p-Limk1-T508/p-Cofilin regulatory pathway, as reported in the Nogo66/ NgR interaction by Ohashi et al.[41]. The signaling pathway triggered

in neuronal cells treated with ePgk1-incubation plus Eno2 transfection presented in this study is consistent with that triggered by ePgk1-incubation alone, as previously reported[16]. Therefore, we conclude that membranous Eno2 on neuronal cells serves as a receptor that interacts with ePgk1.

Hafner et al.[34] incubated human brain Eno2 with neuroblastoma SH-SY5Y cells. They reported that this exogenous synthetic peptide could promote neurite outgrowth through activation of both PI3K/Akt and MAPK/ERK signaling pathways. However, unlike the signaling pathways involved in Eno2-stimulated neurite outgrowth proposed by Hafner et al.[34], both ePgk1-incubation alone and Eno2/ePgk1 interaction demonstrated by Lin et al.[16] and this study, respectively, trigger the reduction of p-Cofilin through decreasing the p-P38-T180/p-MK2-T334/p-Limk1-S323 signaling pathway, which, in turn, enhances the neurite outgrowth of motor neurons. To account for the inconsistency between these studies, Hafner et al.[34] based their study on the internalization of exogenously incubated synthetic peptide into cells, while the studies of Lin et al.[16] and this report are based on the interaction between two intact proteins: membranous receptor Eno2 and ligand ePgk1.

Compared to the control IgG group normalized as 1 in the first experiment, we noticed that the levels of p-Cofilin, p-Limk1-S323, and p-P38 were reduced by 10%, 1%, and 11%, respectively, in cells incubated with Pgk1 plus Eno2 antibody (Fig. 7c, d). These results indicate that p-Cofilin and p-P38 were slightly reduced in cells in spite of blocking Eno2 receptor by Eno2 antibody. Meanwhile, compared with control siRNA-transfected cells in the parallel siRNA-knockdown experiment, the reduction of p-Cofilin, p-Limk1-S323, and p-P38, as mediated by ePgk1, was almost abolished in cells incubated with Pgk1 plus Eno2-siRNA transfection (Supplementary Fig. 7a, b). Yet, we also noticed that p-Cofilin could only be reduced by 7%. Based on this evidence from both experiments, it can be concluded that the reduction of p-Cofilin, as mediated by ePgk1, is nearly, but not completely, abolished in the absence of Eno2, further supporting our hypothesis that ePgk1 ligand triggers the signaling pathway that causes neurite outgrowth by interacting with receptor Eno2 on the neural membrane. Consequently, the interaction between ePgk1 and Eno2 plays an important role in reducing p-Cofilin, even though it suffers only a slight reduction when Eno2 receptor is blocked or knocked down. Based on these results, it is plausible that the dosage of either Eno2-siRNA or Eno2 antibody employed in this study could not completely eliminate the presence of endogenous Eno2 in the treated cells. In fact, western blot analysis demonstrated that Eno2 extracted from the Eno2-siRNA-transfected NSC34 cells was reduced by only 80% (Supplementary Fig. 3a), notwithstanding that Eno2-siRNA is an effective blocker of Eno2 expression. This, in turn, increases the speculation that ePgk1 interacts with Eno2 receptor remaining, even after blockage or knockdown, and that this remainder is sufficient for the reduction in p-Cofilin. It is also possible that ePgk1 affects the p-P38-T180/p-Limk1-S323/p-Cofilin pathway in a manner that departs from its major interaction with intracellular Eno2. In this scenario, additive ePgk1 could bind some minor, or alternative, as-yet-unidentified receptor, or it could be transported intracellularly through endocytosis, to reduce p-Cofilin, in turn slightly favoring axonal branching. Results from the branched neurite experiment might support this hypothesis since the percentage of branched neurites that occurred in embryos injected with eno2-MO plus Pgk1 was slightly higher than that of embryos injected with eno2-MO alone, but lower than that of embryos injected with ePgk1 alone (Supplementary Fig. 5b–d). However, only further study will confirm the above hypotheses.

Finally, we extracted total proteins from NSC34 cells, either incubated with Eno2 antibody or transfected with Eno2-siRNA,

and analyzed the protein levels involved in the p-P38/p-Limk1/p-Cofilin pathway. Compared with control IgG cells, we found that the protein level of the p-P38/p-Limk1-S323/p-Cofilin axis was increased to 113%, 77%, and 62%, respectively, in Eno2-antibody-treated cells (Fig. 7c, d). Similarly, the protein levels of p-P38/p-Limk1-S323/p-Cofilin were all significantly increased to 84%, 59%, and 54%, respectively, in cells incubated with Eno2-siRNA alone (Supplementary Fig. 7a, b). Based on these results, it could be speculated that intracellular Eno2 per se may repress the p-P38-T180/p-Limk1-S323/p-Cofilin pathway through some unknown regulatory mechanism apart from its interaction with ePgk1. Therefore, during the Eno2-knockdown experiment, it is plausible that the remaining unsilenced Eno2 might slightly attenuate p-Cofilin in a manner independent of its interaction with ePgk1, giving still a third explanation for the effect of ePgk1 on neurite outgrowth in cells lacking the Eno2 receptor.

## Methods

**Cell culture.** Cell line HEK293T provided by Prof. Hsi-Yuan Yang, National Taiwan University, were maintained in DMEM high glucose (Gibco) plus 10% Fetal Bovine Serum (Gibco) and 1% penicillin/streptomycin (Gibco) at 37 °C under 5% $CO_2$. Cell line NSC34, the motor neuron derived from Nuroblastoma × Spinal Cord (NSC) hybrid cell lines[42], provided by Dr. Neil Cashman, University of Toronto, was maintained in DMEM high glucose plus 10% fetal bovine serum and 1% penicillin/streptomycin at 37 °C under 5% $CO_2$. Insect cell line Sf21 provided by Prof. Yen-Ling Song, National Taiwan University, were maintained in Grace's Insect Medium (Gibco) plus 10% Fetal Bovine Serum at 28 °C under 5% $CO_2$.

**Cell transfection.** HEK293T and NSC34 cells were sub-cultured in the six-well-dish starting with a density of about $1 \times 10^5$ cells each well. Transfection was performed if cells grew to about 60–70% fullness. First, either DNA or siRNA was mixed with Opti-MEM medium (Gibco) without serum to make the final volume of 250 μL. Second, 7 μL Lipofectamine 2000 Transfection Reagent (Thermo Fisher Scientific) was taken to add into 243 μL Opti-MEM medium, then they were mixed with the above DNA or siRNA at room temperature for 20 min. Third, after removing the original culture medium in the six-well-dish, we added 1 mL of 1×PBS to rinse for twice. Fourth, 500 μL of Opti-MEM medium were slowly added into the well, then 500 μL of the mixture was added at 37 °C. Fifth, after 4-h incubation, the mixture in the well was removed and added 2 mL DMEM medium containing 10% FBS and 1×penicillin/streptomycin at 37 °C.

For the transfection of insect cell Sf21, the Baculovirus protein expressed plasmid was co-transfected with BacPAK™6 DNA (TaKaRa) into the Sf21 cells. After 4-day culturing, the viral suspension carrying the Baculovirus-expressing protein was collected. Then, we took 1 ml of this viral suspension to add into a 25-$cm^2$ cell culture dish and cultured for 4 days. Finally, the floating Sf21 cells infected with Baculovirus expression protein were collected for protein extraction.

**Antibody treatment.** NSC34 cells were seeded in DMEM high glucose (Gibco), 10% fetal bovine serum (FBS) (Gibco) and 1% penicillin/streptomycin (Gibco) medium onto a six-well plate at cell density of $3 \times 10^4$ per well. After 24 h incubation, culture medium was replaced by 2.5% FBS DMEM medium to stimulate differentiation of NSC34. After another 24 h of incubation, either normal mouse IgG (RRID:AB_73718) as control or 4 μg Eno2 antibody (NSE-P1) (RRID:AB_627513) were added. Cells were then harvested after treatment for 24 h.

**Zebrafish.** Zebrafish (Danio rerio) wild-type AB strain (RRID:ZIRC_ZL1) and transgenic line Tg(mnx1:GFP) (RRID:ZIRC_ZL1163)[26] were purchased from ZIRC. The experiments and treatments of this zebrafish model have been reviewed and approved by the Fu Jen Catholic University and MacKay Medical College Institutional Animal Care and Use Committee with ethics approval numbers A11064 and A1050012, respectively.

**Gene cloning.** NSC34 cells and zebrafish embryos at 24 h post-fertilization (hpf) were homogenized with TRIzol Reagent (Invitrogen) to extract total RNAs according to the manufacturer's instructions. The first strand of cDNA was synthesized from 2 ng of total RNAs using SuperScript III (Invitrogen). Then, RT-PCR was performed to clone the cDNAs of coding sequences of mouse Pgk1, TRPC5, Tlr9, mouse Enolase-1 (Eno1), Eno2 and Eno3 from the cDNA of NSC34 and C2C12 cells and zebrafish Pgk1 (zPgk1), zebrafish enolase-1 (eno1), eno2 and eno3 from the cDNA of zebrafish embryos using each set of forward and reverse primers listed in Supplementary Table 2. Thirty-two cycles of PCR amplification were performed by HiFi-DNA polymerase (KAPA). Each cycle consisted of denaturation for 30 s at 94 °C, 1 min of annealing at 57 °C, and 30 s of extension at 72 °C. The last extension step was extended for 10 min at 72 °C. All PCR fragments were

ligated with pGEM-T Easy vector (Promega), transformed into *Escherichia coli* DH5α, and confirmed by sequencing.

**Plasmid constructs**. The full-length coding region of Pgk1 and zPgk1 cDNA fused with reporter Flag peptide was inserted into pCS2 to generate 5.3-kb pCS2-Pgk1-Flag and 5.3-kb pCS2-zPgk1-Flag, respectively. Different deletion clones of Pgk1 cDNA, such as Pgk1(1–145), Pgk1(146–417), Pgk1(225–417), and Pgk1(325–417), were cloned by PCR using each set of primers listed in Supplementary Table 2. Thirty cycles of PCR amplification were performed by HiFi-DNA polymerase (KAPA). Each cycle consisted of denaturation for 30 s at 94 °C, 1 min of annealing at 55°C, and 30 s of extension at 72°C. The last extension step was extended for 10 min at 72 °C. Various PCR-amplified fragments were individually inserted into pCS2 to generate pCS2-Pgk1(1–145)-Flag (4.5 kb), pCS2-Pgk1(146–417)-Flag (4.9 kb), pCS2-Pgk1(225–417)-Flag (4.7 kb) and pCS2-Pgk1(325–417)-Flag (4.4 kb). In addition, the coding sequences of Eno1, Eno2 and Eno3 cDNAs fused with Myc cDNA were inserted into pCS2 to generate pCS2-Eno1-Myc (5.4 kb), pCS2-Eno2-Myc (5.4 kb) and pCS2-Eno3-Myc (5.4 kb), respectively.

Two deletion clones of Eno2 cDNA, Eno2(Δ405–431) and Eno2(Δ432–434), were cloned by PCR using each set of primers listed in Supplementary Table 2. Thirty cycles of PCR amplification were performed by HiFi-DNA polymerase. Each cycle consisted of denaturation for 30 s at 94 °C, 1 min of annealing at 55 °C, and 50 s of extension at 72 °C. The last extension step was extended for 10 min at 72 °C. These two PCR-amplified fragments were individually inserted into pCS2 to generate pCS2-Eno2(Δ405–431)-Myc (5.3 kb) and pCS2-Eno2(Δ432–434)-Myc (5.4 kb), respectively.

Plasmids used for the Baculovirus protein expression system were constructed by PCR to amplify the DNA fragments of Pgk1-Flag, Pgk1(1–145)-Flag, Pgk1(146–417)-Flag, Pgk1(225–417)-Flag, Pgk1(325–417)-Flag, TRPC5-Myc, Tlr9-Myc, Eno1-Myc, Eno2-Myc, Eno2(Δ405–431)-Myc, Eno2(Δ432–434)-Myc and zPgk1-Flag, using each set of forward and reverse primers listed in Supplementary Table 2. All PCR-amplified fragments were inserted individually into pVL1392 (BD) to generate pVL-Pgk1-Flag (10.8 kb), pVL-Pgk1(1–145)-Flag (10 kb), pVL-Pgk1(146–417)-Flag (10.4 kb), pVL-Pgk1(225–417)-Flag (10.2 kb), pVL-Pgk1(325–417)-Flag (9.9 kb), pVL-TRPC5-Myc (12.5 kb), pVL-Tlr9-Myc (12.7 kb), pVL-Eno1-Myc(10.9 kb), pVL-Eno2-Myc (10.9 kb), pVL-Eno2(Δ405–431)-Myc (10.8 kb), pVL-Eno2(Δ432–434)-Myc (10.9 kb) and pVL-zPgk1-Flag (10.8 kb).

Plasmids used for the overexpression of zebrafish enolases in zebrafish embryos were constructed by PCR to amplify the DNA fragments of zebrafish eno1, eno2, and eno3, using each set of forward and reverse primers listed in Supplementary Table 2. All PCR-amplified fragments were inserted individually into pCS2-Myc to generate pCS2-eno1-Myc (5.4 kb), pCS2-eno2-Myc (5.4 kb), and pCS2-eno3-Myc (5.4 kb).

**Purification of recombinant Pgk1 protein fused with Flag**. Recombinant protein of Pgk1 fused with Flag, such as pCS2-Pgk1-Flag, pCS2-Pgk1(1–145)-Flag, pCS2-Pgk1(146–417)-Flag, pCS2-Pgk1(225–417)-Flag, pCS2-Pgk1(325–417)-Flag and pCS2-zPgk1-Flag, were individually transfected into HEK293T cells. In parallel, plasmids of pVL-Pgk1-Flag, pVL-Pgk1(1–145)-Flag, pVL-Pgk1(146–417)-Flag, pVL-Pgk1(225–417)-Flag, pVL-Pgk1(325–417)-Flag and pVL-zPgk1-Flag were transfected into Sf21 insect cells. All transfected cells were lysed using Pierce IP lysis buffer (Thermo Fisher Scientific) with protease inhibitor cocktail (Roche). After cell debris was removed by centrifugation, anti-Flag beads (Sigma-Aldrich) were added to cell extracts and incubated at 4 °C for 16 h. Pgk1-fused proteins were eluted by incubation with 3× Flag peptide (Sigma-Aldrich) for 1 h. The eluted protein was kept for later experiments.

**Flag pull-down assay**. After recombinant Pgk1-Flag protein was purified, immunoprecipitation (IP) of Pgk1-Flag with membrane protein extracts isolated from NSC34 cells followed the protocol described by the Pierce Crosslink Immunoprecipitation Kit (Thermo Fisher Scientific). The resultant IP products were analyzed by SDS-PAGE on a 10% gel after silver staining.

**In-gel digestion and LC-MS/MS analysis**. The IP products shown on SDS-PAGE after silver staining were cut into many gel slices about 1 mm in length. Then, these gel slices were applied to perform in-gel digestion and LC-MS/MS, using the Mascot Distiller (Matrix Science). The resultant MGF file was searched using the Mascot Search Engine (v2.2, Matrix Science) under the following conditions: (i) protein database set as Swiss-Prot; (ii) taxonomy set as Homo sapiens; (iii) one trypsin missed cleavage allowed; (iv) peptide mass tolerance set at ±0.5 Da and fragment mass tolerance set at ±0.5 Da; (v) Carbamidomethyl (Cys) chosen as a fixed modification; and (vi) Oxidation (Met) and deamidation (Asn and Gln) chosen as variable modifications.

**IEM**. NSC34 cells were cultured on a 30-mm dish at a density of $4 \times 10^5$ cells per dish and incubated in medium containing 500 ng/μl of either recombinant Pgk1-Flag or EGFP-Flag (control) for 24 h at 37 °C. Cell pellets were collected by centrifugation (2000 rpm, 1 min) and fixed with 2.5% glutaraldehyde/3% formaldehyde for 2 h. After fixation, cells were washed by 0.1 M phosphate buffer for 1 h, followed by dehydration with ascending alcohol concentrations (30%, 50%, 70%, 80%, 90%, 100%; each step for 15 min). Dehydrated cell pellets were infiltrated with and

embedded in LR White resin (London resin) for 24 h and then polymerized in an oven at 60 °C for 2 days. Ultrathin sections (90 nm) were cut by ULTRACUT R (LEICA) and Diamond Knives (DiATOME). The resultant ultrathin sections were collected on Formvar/Carbon 100 mesh NI grids. For the double immune-labeling experiment, samples were blocked in 2.5% BSA (Sigma-Aldrich) for 1 h and incubated in antisera against Eno2 (NSE-P1) (RRID: AB_627513; 6 μg/30 μl) and Flag peptide (RRID: AB_446355; 6 μg/30 μl) overnight at 4 °C, followed by washing with 1× TBS for 1 h. After washing, the grids were incubated in a 12-nm colloidal gold donkey anti-mouse IgG (RRID: AB_2340822; 1:20) and an 18-nm colloidal gold goat anti-rabbit IgG (RRID: AB_2338017; 1:20) for 2 h at room temperature, followed by washing with 1X TBS for 1 h and staining with 2% uranyl acetate for 20 min. Specimens were examined under the HITACHI H-7650 electron microscope.

**Cell surface crosslinking-IP**. After 48 h, insect cells were infected with pVL-TRPC5-Myc, pVL-Tlr9-Myc, and pVL-Eno1-Myc, pVL-Eno2-Myc, pVL-Eno2(Δ405–431)-Myc, and pVL-Eno2(Δ432–434)-Myc. The recombinant protein of Pgk1-Flag, Pgk1(1–145)-Flag, Pgk1(146–417)-Flag, Pgk1(225–417)-Flag or Pgk1(325–417)-Flag was then added into the insect cell culture medium and incubated for 2 h, followed by mixing with bis[sulfosuccinimidyl] suberate (BS3) to perform crosslinking-IP according to the procedures described for BS3 Cross-linkers (Thermo Fisher Scientific). Cell lysates were then subjected to IP with either anti-Flag (Sigma-Aldrich) or anti-Myc beads (Sigma-Aldrich). The IP products were then prepared for western blot analysis with anti-Flag (RRID:AB_446355; 1:10,000) and anti-Myc (RRID:AB_439694; 1:1000). To carry out the antibody-blocking experiment, we added antibody against Eno2 (NSE-P1) (RRID: AB_627513) into the insect cell culture medium at a concentration of 2 μg/ml and incubated for 1 h before performing Crosslinking-IP.

**Western blot analysis**. Total proteins extracted from NSC34 cells and zebrafish embryos at 48 hpf were lysed by whole-cell extract buffer[43] containing cOmplete™, EDTA-free Protease Inhibitor Cocktail (Sigma-Aldrich) and PhosSTOP (Sigma-Aldrich). Lysates derived from the same protein source were dispensed for multiple groups of loading to carry out SDS-PAGE on a 10% gel simultaneously. After electrophoresis, we separately performed western blot analysis for each group of loading using one specific antiserum. Antibodies against phosphorylated Cofilin at S3 (RRID:AB_2080597; 1:1000), Cofilin (RRID:AB_10622000; 1:1000), phosphorylated Limk1 at S323 (RRID:AB_2136940; 1:1000), phosphorylated Limk1 at T508 (RRID:AB_2136943; 1:500), LIM domain kinase 1 (Limk1) (RRID:AB_648350; 1:500), phosphorylated P38 at T180 (RRID:AB_331641; 1:1000), P38 mitogen-activated protein kinases (P38) (RRID:AB_330713; 1:1000), Eno1 (RRID:AB_1118874; 1:1000), Eno2 (NSE-P1) (RRID:AB_627513; 1:1000), eno2 (SC06-28; 1:1000), goat anti-mouse-HRP (RRID:AB_955439; 1:5000), goat anti-rabbit-HRP (RRID:AB_631746; 1:5000) and chicken anti-goat-HRP (RRID: AB_639230; 1:5000) were used. Antibody against α-tubulin (RRID:AB_477579; 1:1000) served as an internal control. Band intensities shown on western blotting were visualized by enhanced chemiluminescence (Millipore) and quantified by ImageJ software. The relative intensity of examined protein among all treatments compared to that of internal loading control (tubulin) was measured. In parallel, the relative intensity obtained from the pCS2-vector (control group) against internal tubulin was normalized as 1. Based on the control group set as 1, we compared all the fold changes of examined protein among all treatments in same group, which was horizontally indicated on the blot.

**Zebrafish husbandry and microscopy observation**. Production and stage identification of zebrafish embryos followed the description in ref. [16]. Fluorescence was visualized with a fluorescence stereomicroscope (Leica) and a confocal spectral microscope (ZEISS).

**Knockdown of Eno1 and Eno2 by small interfering RNA (siRNA) in cell line**. Eno1-siRNA (GCACAGAGAAUAAAUCUAATT) and Eno2-siRNA (GAUC-CUUCCCGAUACAUCATT) were designed to specifically knock down Eno1 and Eno2 through base-pairing with corresponding nucleotides from 294 to 310 nt at Exon 5 of Eno1 mRNA and from 796 to 814 nt at Exon 7 of Eno2 mRNA, respectively. The Silencer™ Select Negative Control No. 1 siRNA, designed to validate non-silencing siRNA, which served as a negative control, was purchased from Thermo Fisher Scientific. All siRNAs were prepared from a stock concentration of 5 nmol and diluted to the desired concentration for transfection of NSC34 cells. After 24-h transfection, total proteins extracted from transfected cells were analyzed by western blot using antibody against Eno1 or Eno2. Data are presented independently in triplicate.

**Knockdown of zebrafish eno2 in embryos by antisense morpholino oligonucleotides (MO)**. The eno2-MO (GGCAATGATGCTTACAACAGACATC), designed to specifically knock down eno2 through base-pairing with corresponding nucleotides of eno2 mRNA from −1 to 24 nt, and negative control MO (eno2-5mis-MO), designed as GGCATTCATCCTTACAAGACACATC (mismatched nucleotides are underlined), were purchased from Gene Tools (USA). All MOs were prepared at a stock concentration of 1 mM and diluted to the desired concentration for injection into transgenic line Tg(mnx:GFP) at one-celled stage. To study neurite outgrowth of zebrafish embryos derived from transgenic line Tg(mnx:GFP), we followed the protocols described in refs. [16,44,45]. After

microinjection, we observed the axonal development of CaP motor neuron at 30 hpf under fluorescence microscopy. The number of branched axons that occurred in CaP motor neurons between 6 and 20 somites was counted. Axonal phenotype of CaP motor neurons was defined as motor axons that displayed at least one branched axon shown on both sides of embryos. Data were presented independently in triplicate as the average of 45 embryos.

**Overexpression of zebrafish eno2 and Pgk1 treatment in zebrafish embryos**. Capped mRNAs encoding zebrafish eno1, eno2, and eno3 were synthesized according to the protocol of the mMESSAGE mMACHINE™ SP6 Transcription Kit (Thermo Fisher Scientific AM1340). The resultant mRNAs were diluted to 176 ng/µl with sterilized double distilled water. Approximately 2.3 nl were injected into transgenic line *Tg(mnx:GFP)* at one-celled stage. After microinjection, we observed the axonal development of CaP motor neurons at 30 hpf under fluorescence microscopy. We defined the phenotypes as follows: (a) normal axonal phenotype neurons: the CaP of primary motor neurons (PMN) located at the 10th–20th somites developed normally; and (b) branched axonal phenotype neurons: the CaP of PMN exhibited at least one neurite among an examined range having an ectopic branch shown on the 30-hpf embryos.

For the incubation of recombinant zPgk1 in zebrafish embryos, we first purified the recombinant fusion protein zPgk1-Flag produced by HEK293T cells. The chorion of embryos at 6 hpf was partially removed by sharp forceps (Dumont No. 5), if necessary, followed by their immersion in embryo media containing zPgk1-Flag with a concentration of 300 ng/ml and incubation for 24 h at 26–28 °C. Then, embryos at 30 hpf were fixed with 4% PFA overnight, and their neurite outgrowth of motor neurons was observed under fluorescence microscopy.

**In vitro platform used to study neurite outgrowth of motor neurons derived from NSC34 cells**. NSC34 cells were seeded in 10% FBS DMEM medium into wells of a six-well plate at cell density of $1 \times 10^5$ per well. After 24-h incubation, cells were transfected with pCS2, pCS2-Eno1, pCS2-Eno2 and pCS2-Eno3 individually. After transfection for four hours, culture medium was replaced by 1% FBS DMEM medium (Serum deprivation medium, SDM) to stimulate differentiation of NSC34. After 24-h incubation, medium was refreshed by SDM, including 66 ng/ml mPgk1-Flag. After 48-h incubation, the neurites extending from each NSC34 cell were captured under phase contrast microscopy (ZEISS Axio Observer Z1, Göttingen, Germany) at 10 times magnification. Five experimental groups were established, as follows: cells treated with pCS2 transfection, pCS2 transfection plus Pgk1 incubation, pCS2-Eno1 transfection plus Pgk1 incubation, pCS2-Eno2 transfection plus Pgk1 incubation and pCS2-Eno3 transfection plus Pgk1 incubation. For each trial, 15 pictures were randomly taken for each experimental group, followed by analysis of neurite cells and neurite length using the *Neurite Outgrowth Analysis Application* Module of MetaMorph software. Cells with axons longer than 30 µm were defined as neurite-bearing cells. The percentage of neurite-bearing cells among examined cells and the average length of each neurite were calculated. Datum from each trial was averaged from 15 photos. Final data were averaged from three independent experiments and expressed as mean ± SD.

**Statistics and reproducibility**. All the experiments were performed in triplicates and data were averaged from three independent experiments and presented as mean ± SD. One-way ANOVA, followed by Tukey's multiple comparison test, or Student's *t* test for comparisons was used to perform statistical analysis. Significance was determined at *P* value as indicated in the figure legends. NS: no significance at $P > 0.05$; *($P < 0.05$), **($P < 0.005$), and ***($P < 0.001$): statistically significant at the level as indicated.

**Reporting summary**. Further information on research design is available in the Nature Portfolio Reporting Summary linked to this article.

## Data availability
The authors declare that all data supporting the findings of this study are available within the paper and/or the Supplementary Information. The unedited/uncropped western blot gels are included in Supplementary Figs. 8–18. The source data behind the graphs in the paper are included in Supplementary Data 1. All other data are available from the corresponding authors upon reasonable request.

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

## Acknowledgements

This work was supported by the National Science and Technology Council, Taiwan, ROC (MOST 111-2313-B-030-001-), partially supported by MacKay Medical College, Taiwan (1091B12), and partially supported by the Liver Disease Prevention and Treatment Research Foundation, Taiwan. We were grateful to Prof. In-Lung Chu and Dean Ping-Keung Yip, School of Medicine, Fu Jen Catholic University, Taiwan, for the financial support of building culture room and wet lab. We were also grateful to the technical staff members Yi-Chun Chuang and Pei-Yin Wu, Technology Commons, College of Life Science, NTU, for helping with confocal laser scanning microscopy and transmission electron microscopy.

## Author contributions

C.-Y.F. conducted the main experiments, such as screening, recombinant proteins preparation and cell line experiments, while H.-Y.C. performed transgenic fish experiments and microarray. C.-Y.F., H.-Y.C., and C.-Y.L. prepared plasmids, proteins, cell lines, and transgenic fish. S.-J.C. guided IEM technology and provided material resources. J.-C.S. and H.-J.T. conceptualized and directed the project. H.-J.T. reorganized and edited the draft written by C.-Y.F.

## Competing interests

The authors declare no competing interests.
