## [Peer Review File · Communications Biology]

Reviewers' comments:

Reviewer #1 (Remarks to the Author):

In ALS patients, overexpression of NogoA in muscle cells is correlated to the loss of motor endplates. Notably, NogoA-overexpressing skeletal muscle cells show reduced Pgk1 secretion thereby increasing p-Cofilin, a growth cone collapse marker, by triggering p-P38-T180/p-Limk1-S323 signals in motoneurons and attenuating the neurite outgrowth of motoneurons. This is an interesting and important paper in which the missing membrane receptor for the extracellular Pgk1 is identified as Eno2 through a variety of experimental approaches, combining biochemical and cell biological approaches as well as assessing neurite branching in transgenic zebra fish with motor neurons labeled by GFP. Several modifications will further strengthen the manuscript. Overall, however, this manuscript deserves a publication in Nature Com, as it will be widely impacting in the area of ALS and neurobiology.

1) For Fig 5A, the images are not convincing at all. They should show much wider field of images to support the quantification data in 5B.

2) Some descriptions of the figures and text are quite confusing and need to be clarified. Why stress about the fact that the 404-434 amino acid region of Eno2 is hydrophilic? Why is this important for potential interactions with ePgk1? Similarly why bother with PDZ? No PDZ domain seems to be present in ePgk1. Also the nomenclature for this region in Fig 3A as 'promoter neuronal domain' is confusing. I think the authors meant to say a domain that promotes neuronal survival??? Finally, Fig. 6A, the nomenclature for wt-like neurons vs. branched neurons is confusing. It sounds like branched neurons are NOT wt-like.

3) For Fig. 4, the authors stated: "Since the domain including 325th-417th amino acid residues (325-417 domain) of ePgk1 were shared by all examined truncated forms displaying positive reaction, we concluded that the 325-417 domain of ePgk1 was the key domain interacting with Eno2." They should consider the possibility for another independent Eno2-interacting domain in the amino acids 146-325.

4) For all immunoblotting analysis (e.g., Fig 5D, 6D) as well as other experimental results, the reproducibility information is not found in either figure legends or text. The authors should provide this information, including how many times each assays were done, etc.

5) S-Fig. 4 has no information on Eno2 protein level reduction as opposed to the statement in the text. The authors should carefully go through all the figures and text.

Reviewer #2 (Remarks to the Author):

In this study, Fu and colleagues set out to identify the neuronal receptor for extracellular phosphoglycerate kinase (ePgk1) that was previously shown to stimulate neurite outgrowth. Besides its canonical role in glycolysis, phosphoglycerate kinase can be secreted. Work from this team had demonstrated that ePgk1 released by muscle cells promotes motor neuron outgrowth by suppressing a classic intracellular cell-repulsion pathway (p38/LIMK/Cofilin).

In this study, the Authors searched for ePgk1 interactors through MS proteomics and identified Enolase-2 (Eno2) as candidate receptor. ePgk1/Eno2 association at the cell surface was inferred through biochemical assays and double immunogold EM detection. In vitro pull-down with truncated mutants revealed the amino-acid sequences required for interaction. ePgk1/Eno2 (but not Eno1 or Eno3) increased neurite outgrowth in a motor neuron-like cell line (NSC34). Zebrafish embryos overexpressing Eno2 and incubated with ePgk1 displayed ectopic branching of motor axons and

reduced p-Cofilin levels while Eno2 knockdown had opposite effects. The intracellular pathway regulated by ePkg1/Eno2 was same that the Authors had described in their previous work, involving p38/LIMK/Cofilin. Finally, an Eno2-blocking antibody that interferes with ePkg1/Eno2 interaction affected the signaling outcome.

The findings are interesting as they provide insight into the noncanonical extracellular function of Pkg1 and might have implications for interpreting disease conditions in which the levels of this protein are altered. The study has however limitations that should be addressed either experimentally or with discussion.

(1) It is unclear whether the topography of Eno2 at the plasma membrane is compatible with its proposed role as ePkg1 receptor. A number of previous studies (cited) have found Eno1 and Eno2 associated with cellular membranes. However, in order to function as a receptor the way the Authors suggest in their model (Figure 8), the amino-acid sequence that they have identified as required for ePkg1 binding should be exposed to the extracellular space. Are there evidences for this configuration? The model in Fig. 8 suggests a transmembrane topography that is not supported by the data. This should be addressed.

(2) NSC34 cells are useful for biochemical analysis but are not necessarily the best system to study axonal outgrowth. Primary neuronal cultures would be recommended. I appreciate this might require setting up assays that are currently unavailable in the Team's laboratories. Zebrafish embryos are used here as an in vivo model to study the effects of ePkg1/Eno2 on neurite extension. However, I am not convinced that the reported phenotypes (ectopic axon branching) are an indication of "increased neurite outgrowth of motor neurons". They seem more in line with a generic effect on cytoskeletal remodeling and possibly a perturbation of normal axon growth. In the absence of a clear readout for axon elongation, I would recommend the Authors are cautious with the interpretation of the phenotypes. The cellular and in vivo experiments clearly show that ePkg1/Eno2 impinges on a classic axis for cytoskeletal regulation. Whether axon outgrowth is truly the main physiological outcome of the signaling I think remains to be demonstrated.

(3) I was not sure how to interpret the effect of Eno2 blocking antibody (Fig. 7) and siRNA (Suppl. Fig. 4) in the absence of ePkg1 ligand. Can the Author provide an explanation?

(4) Several points in the manuscript require attention:

- lines 269-276: the effect on p-Cofilin in NSC34 cells has already been presented for Suppl. Fig. 1. The two paragraphs could be combined to avoid redundancy.

- line 279: "decreased in the culture medium". Is this a typo for "total protein lysate"?

- lines 305-312 describe the efficiency of siRNA already presented for Suppl. Fig. 1. Could be removed to avoid redundancy.

- line 314: one of the conditions should be si-eno2 only (without ePkg1)

- I recommend shortening and streamlining the Discussion to avoid diluting the interpretation of the findings.

What is the relevance of the section on Enolase family (line 347).

Other points:

- Suppl Fig 1: mark specific bands on WB (A,B)

- For how long were embryos incubated with recombinant ePkg1? I could not find in the Methods. It

should be added to the legend.

- What are the levels of Eno2 protein overexpression achieved after mRNA injection in embryos (Figure 6)?

- Suppl. Figure 2 examines motor neuron phenotypes in eno2-MO embryos. The levels of p-cofilin could be measured as in Figure 6.

Reviewer #3 (Remarks to the Author):

Overexpression of NogoA in muscle cells is known to positively correlate with the loss of motor endplates in diseases such as Amyotrophic Lateral Sclerosis. NogoA-overexpressing skeletal muscle cells have the capacity to reduce secretion of phosphoglycerate kinase 1 (Pgk1), thus increasing p-Cofilin, a growth cone collapse marker, by triggering p-P38-T180/p-Limk1-S323 signals in motoneurons and attenuating the neurite outgrowth of motoneurons (NOMN). Thus, the authors investigated whether NOMN could be controlled by extracellular Pgk1 (ePgk1) plus an interacting neural receptor which is not currently known. The authors found that Enolase-2 (Eno2) can exhibit strong affinity with recombinant Pgk1-Flag, and the 325th-417th amino acids of ePgk1 may interact with the 404th-431st amino acids of Eno2. This is convincing. However, the role of Eno1 is still not clear although the authors showed some data suggesting neither Eno1 nor Eno3 enhanced NOMN in NSC34 cells and zebrafish embryos. There may be a minor role of Eno1 since Eno1 has the dominant function on the surface. The function of Eno3 is somewhat supported by the data. However, the role of Eno2 could be dominant as it is strongly supported by the data.

Figure 6 and 7, Western blot data. Some of the blots are not clear. Figure 6 Cofilin data are not convincing. In figure 7, bands with similar molecular weight are presented with a single loading control. This creates confusion.

The authors utilized immunoprecipitation and LC-MS/MS analyses to screen putative membrane receptors of ePgk1. Studies here with the screening and interaction of proteins in cell lines and zebrafish embryos convincingly showed that Eno2 had the strongest affinity with Pgk1, and may regulate the P38/Limk1/Cofilin signaling axis, improving the neurite outgrowth of motor neurons. Data presented here support association of ePgk1 with Eno2 on the cell membrane of neural cells, and the 404th-431st amino acid residues of Eno2 may comprise the key domain that interacts with ePgk1. Data also support the finding that the 325-417 domain of ePgk1 could be the key domain interacting with Eno2. However, the role of Eno1 should be revisited as this reviewer thinks that Eno1 may have a minor role in the interaction with ePgk1.

Referee #1

In ALS patients, overexpression of NogoA in muscle cells is correlated to the loss of motor endplates. Notably, NogoA-overexpressing skeletal muscle cells show reduced Pgk1 secretion thereby increasing p-Cofilin, a growth cone collapse marker, by triggering p-P38-T180/p-Limk1-S323 signals in motoneurons and attenuating the neurite outgrowth of motoneurons. This is an interesting and important paper in which the missing membrane receptor for the extracellular Pgk1 is identified as Eno2 through a variety of experimental approaches, combining biochemical and cell biological approaches as well as assessing neurite branching in transgenic zebra fish with motor neurons labeled by GFP. Several modifications will further strengthen the manuscript. Overall, however, this manuscript deserves a publication in Nature Com, as it will be widely impacting in the area of ALS and neurobiology.

(1) For Fig 5A, the images are not convincing at all. They should show much wider field of images to support the quantification data in 5B.

Author's response to (1):

In response to your comment, we performed a new experiment in order to capture images with a wider field, as suggested, followed by analysis using the *Neurite Outgrowth Analysis Application* Module of MetaMorph software. Accordingly, we have added a revised description in M&M, new Figure 5 (please see below), all 15 original photos taken for each group for one trial (Source data for Figure 5A; see below), and a revised Figure 5 legend, as follows: “(A) Morphology of neurites derived from NSC34 neural cells was observed under microscopy. Five experimental groups were categorized: cells treated with pCS2 transfection, pCS2 transfection plus Pgk1 incubation (pCS2+Pgk1), pCS2-Eno1 transfection plus Pgk1 incubation (pCS2-Eno1+Pgk1), pCS2-Eno2 transfection plus Pgk1 incubation (pCS2-Eno2+Pgk1) and pCS2-Eno3 transfection plus Pgk1 incubation (pCS2-Eno3+Pgk1). Scale bar, 100 μ m. (B) *Neurite Outgrowth Analysis Application* Module of MetaMorph software was used to count neurite-bearing cells and measure neurite lengths. Cell body was marked in white, while neurite was marked in red. (C-D) Statistical analysis: (C) percentage of neurite-bearing cells among examined cells and (D) average neurite length of each group, as indicated. Experiments were performed independently at three trials. Final

data presented as averaged from 3 trials. presented as averaged from 3 trials. presented as averaged from 3 trials. 3 trials.” (Please see Results section from line 1003 of page 45 to line 1014 of page 46)

M&M was revised, as follows: “Neurites extending from each NSC34 cell were captured under phase contrast microscopy (ZEISS Axio Observer Z1, Göttingen, *Germany*) at 10x magnification. Five experimental groups were established, as follows: cells treated with pCS2 transfection, pCS2 transfection plus Pgk1 incubation, pCS2-Eno1 transfection plus Pgk1 incubation, pCS2-Eno2 transfection plus Pgk1 incubation and pCS2-Eno3 transfection plus Pgk1 incubation. For each trial, 15 pictures were randomly taken for each experimental group, followed by analysis of neurite cells and neurite length using the *Neurite Outgrowth Analysis Application* Module of MetaMorph software. Cells with axons longer than 30 μm were defined as neurite-bearing cells. The percentage of neurite-bearing cells among examined cells and the average length of each neurite were calculated. Datum from each trial was averaged from 15 photos. Final data were averaged from three independent experiments and expressed as mean \pm S.D.” (Please see Materials and methods section from line 717 of page 32 to line 728 of page 32)

Fig. 5

Source data for Fig. 5A

(2) Some descriptions of the figures and text are quite confusing and need to be clarified.

(A) Why stress about the fact that the 404-434 amino acid region of Eno2 is hydrophilic?

Author's response to (2A):

We drew this conclusion based on the results from ProtScale-ExPASy software analysis. We added more detailed description in the revised text, as follows: “We used ProtScale-ExPASy software (<https://web.expasy.org/protscale/>) to analyze the hydrophobicity of Eno2 and found the 404th-434th amino acid region located at the C-terminus of Eno2 to be highly hydrophilic (Supplementary Fig. 2B) (below).” We

added this information and a new supplementary figure in the text (Please see Results section from line 148 of page 7 to line 151 of page 7).

Supplementary Fig. 2

(2B) Why is this important for potential interactions with ePkg1?

Author's response to (2B):

To address your question, we added the following revised text: “It has been reported that the C-terminus of Eno2 promotes neuronal survival, differentiation and axonal regeneration (21). Eno1, another membrane protein in the Enolase family, has been reported to interact with plasminogen through its exposure to C-terminal lysine residue (22-23). When we also used ProtScale-ExpASy software to analyze its relative hydrophilicity, we found its C-terminal domain structure to be hydrophilic, similar to that of Eno2 (Supplementary Figs. 2A-B) (below). Extrapolating from this evidence, it was hypothesized that the hydrophilic C-terminus 404th – 434th of Eno2, a peripheral protein located on the cell membrane, might also be a domain that interacts with extracellular ligand Pkg1 we studied here.” (Please see Results section from line 151 of page 7 to line 159 of page 7).

Supplementary Fig. 2

(2C) Similarly why bother with PDZ? No PDZ domain seems to be present in ePkg1.

Author's response to (2C):

It is true that ePkg1 contains no PDZ domain, but this study is concerned with the PDZ domain of receptor Eno2 based on the following reasoning. Our revised explanation is as follows: "First, the C-terminus of Eno2 contains a PDZ-binding motif (432-434: S-V-L) which interacts with other PDZ domain-containing proteins involved in the intracellular redistribution of molecules and signaling pathways (33). For example, PDZ interacts with γ -1-syntrophin to facilitate the transport of Eno2 toward the plasma membrane (33). To determine whether the PDZ-binding motif of Eno2 is

associated with the interaction of eP_{gk1}, we engineered Eno2 with C-terminal deletion and fused with Myc, including Eno2(Δ 432-434)-Myc, which deletes the PDZ-binding motif of Eno2, followed by performing a pull-down assay. As shown in Figure 3B, the P_{gk1}-Flag signal could be pulled down by Eno2-Myc and Eno2(Δ 432-434)-Myc, but not Eno2(Δ 404-431)-Myc, suggesting that the PDZ domain of Eno2 is not involved in the interaction of eP_{gk1} to improve neurite outgrowth.”

(2D) Also the nomenclature for this region in Fig 3A as ‘promoter neuronal domain’ is confusing. I think the authors meant to say a domain that promotes neuronal survival???

Author’s response to (2D):

Thank you for pointing out this mistake. We corrected as you suggested.

(2E) Finally, Fig. 6A, the nomenclature for wt-like neurons vs. branched neurons is confusing. It sounds like branched neurons are NOT wt-like.

Author’s response to (2E):

In response to your comment, we defined these two terms in more detail in the revised Materials and methods to avoid confusion, as follows: “The normal axonal phenotype neurons are the caudal primary (CaP) of primary motor neurons (PMN) located at the 10th -20th somites of 30-hpf embryos (13) from transgenic line Tg(mnx:GFP) (26) with normal development.” (Please see Figure 6A below) “In contrast, branched axonal phenotype neurons, a phenotype of CaP of PMN, exhibited at least one neurite among an examined range having an ectopic branch shown on the 30-hpf embryos.” (Please see Figure 6B below) (Please see Materials and methods section from line 696 of page 31 to line 700 of page 31)

Fig. 6

(3) For Fig. 4, the authors stated: “Since the domain including 325th-417th amino acid residues (325-417 domain) of ePgk1 were shared by all examined truncated forms displaying positive reaction, we concluded that the 325-417 domain of ePgk1 was the key domain interacting with Eno2.” They should consider the possibility for another independent Eno2-interacting domain in the amino acids 146-325.

Author’s response to (3):

In accordance with your suggestion, we designed a new cell-surface crosslinking-immunoprecipitation experiment to determine whether the 146-324 domain of Pgk1 could be possible to interact with Eno2. In this experiment, we subcloned the 146-324 domain from Pgk1 (see upper figure below) and generated its recombinant proteins by the Baculovirus expression system. The figure below compares the control 146-417 domain, which could interact with Eno2, with the 146-324 domain of Pgk1, which could not interact with Eno2. These results suggest that the 325-417 domain of ePgk1 was the key domain interacting with Eno2.

(4) For all immunoblotting analysis (e.g., Fig 5D, 6D) as well as other experimental results, the reproducibility information is not found in either figure legends or text. The authors should provide this information, including how many times each assays were done, etc.

Author's response to (4):

In accordance with your suggestion, we added reproducibility information to the legends of Figures 5D, along with the following text: “(D) Western blot analysis. NSC34 cells were incubated either in the presence (+) or absence (-) of mouse Pdk1 and transfected pCS2-Eno1-, -Eno2- or -Eno3-Myc, followed by analyzing the levels of phosphorylated Cofilin at S3 (p-Cofilin) and total Cofilin. The α -tubulin served as an internal control. Protein levels relative to each internal control set as 1 were presented under each lane. Data were averaged from three independent experiments. (E) Statistical analysis. The change (in fold) of relative intensity of p-Cofilin against α -tubulin of each group was compared to that of the control group which was set as 1. Data were averaged from three independent experiments and presented as mean \pm SD (n = 3). One-way ANOVA, followed by Tukey's multiple comparison test, was used to perform statistical analysis (**, p<0.005).” (Please see legends for Figures 5D from line 1012 of page 46 to line 1023 of page 46)

and 6D, along with the following text: “(D) Western blot analysis. Zebrafish embryos from transgenic line *Tg(mnx1:GFP)* were incubated with (+) or without (-) zebrafish Pdk1 for 24 hr and injected with or without three isoforms of eno mRNAs. Non-injected embryos served as control, followed by collecting total protein lysate from *Tg(mnx1:GFP)* and then analyzing the levels of phosphorylated Cofilin at S3 (p-Cofilin) and total Cofilin. The α -tubulin served as an internal control. Protein levels relative to each internal control set as 1 were presented below each lane. Data were averaged from three independent experiments. (E) Statistical analysis. The change (in fold) of relative intensity of p-Cofilin against internal marker tubulin compared to that of the control group which was set as 1. Data were averaged from three independent experiments and presented as mean \pm SD (n = 3). One-way ANOVA, followed by Tukey's multiple comparison test, was used to perform statistical analysis (**, p<0.005).” (Please see legends for Figures 6D from line 1038 of page 48 to line 1049 of page 48)

(5) S-Fig. 4 has no information on Eno2 protein level reduction as opposed to the statement in the text. The authors should carefully go through all the figures and text.

Author's response to (5):

Thank you for pointing out this mistake. You are right, S-Fig. 4 didn't have

information on Eno2 protein level reduction. We corrected the error and added the following revised text: “compared with cells transfected with control siRNA, the protein levels of p-P38-T180, p-Limk1-S323 and p-Cofilin were all significantly reduced by 58, 34 and 45%, respectively, while the total protein levels of P38, Limk1, and Cofilin, as well as p-Limk1-T508, remained nearly unchanged in cells incubated with Pgk1 and control siRNA (Supplementary Fig. 6A-B), suggesting that the reduction of p-P38/p-Limk1-S323/p-Cofilin axis by ePgk1 was functional in the presence of control siRNA. Moreover, we found that the levels of the above phosphorylated could be restored and total proteins remained unchanged in cells incubated with Pgk1 plus *eno2*-siRNA except that p-Cofilin was reduced by 7% and p-P38 was increased by 14% (Supplementary Fig. 6A-B), suggesting that the reduction of p-Cofilin by ePgk1 was nearly abolished by the presence of *eno2*-siRNA. These results were consistent with those obtained from cells treated with Eno2-antibody, as described above. Meanwhile, when we extracted total proteins from NSC34 cells incubated with *eno2*-siRNA alone, we found that the protein levels of p-P38-T180, p-Limk1-S323 and p-Cofilin were all significantly increased by 84, 59 and 54%, respectively, compared to those of control siRNA-transfected cells.” (Please see Results section from line 330 of page 15 to line 344 of page 16)

Referee #2

In this study, Fu and colleagues set out to identify the neuronal receptor for extracellular phosphoglycerate kinase (ePgk1) that was previously shown to stimulate neurite outgrowth. Besides its canonical role in glycolysis, phosphoglycerate kinase can be secreted. Work from this team had demonstrated that ePgk1 released by muscle cells promotes motor neuron outgrowth by suppressing a classic intracellular cell-repulsion pathway (p38/LIMK/Cofilin).

In this study, the Authors searched for ePgk1 interactors through MS proteomics and identified Enolase-2 (Eno2) as candidate receptor. ePgk1/Eno2 association at the cell surface was inferred through biochemical assays and double immunogold EM detection. In vitro pull-down with truncated mutants revealed the amino-acid sequences required for interaction. ePgk1/Eno2 (but not Eno1 or Eno3) increased neurite outgrowth in a motor neuron-like cell line (NSC34). Zebrafish embryos overexpressing Eno2 and incubated with ePgk1 displayed ectopic branching of motor axons and reduced p-Cofilin levels while Eno2 knockdown had opposite effects. The intracellular pathway regulated by ePgk1/Eno2 was same that the Authors had described in their previous work, involving p38/LIMK/Cofilin. Finally, an Eno2-blocking antibody that interferes with ePgk1/Eno2 interaction affected the signaling outcome.

The findings are interesting as they provide insight into the noncanonical extracellular function of Pgk1 and might have implications for interpreting disease conditions in which the levels of this protein are altered. The study has however limitations that should be addressed either experimentally or with discussion.

(1) It is unclear whether the topography of Eno2 at the plasma membrane is compatible with its proposed role as ePgk1 receptor. A number of previous studies (cited) have found Eno1 and Eno2 associated with cellular membranes. However, in order to function as a receptor the way the Authors suggest in their model (Figure 8), the amino-acid sequence that they have identified as required for ePgk1 binding should be exposed to the extracellular space. Are there evidences for this configuration? The model in Fig. 8 suggests a transmembrane topography that is not supported by the data. This should be addressed.

Author's response to (1):

After first searching for the X-ray structure of Eno2 (PDB ID : 5TD9) from the Protein Data Bank, we presented the structural information using Pymol software (below). As marked in red below, results showed that the 404-431 domain of Eno2 is exposed on the outer membrane protein structure.

Second, we used TMHMM - 2.0 software (<https://services.healthtech.dtu.dk/service.php?TMHMM-2.0>) to analyze the transmembrane property of Eno2. According to the results, Eno2 displays no transmembrane property (Supplementary Fig. 1) (below). We also used ProtScale-ExpASy software (<https://web.expasy.org/protscale>) to analyze the hydrophobicity of Eno2. According to these results, the 404-434 domain at the C-terminus of Eno2 is highly hydrophilic and exposed to the extracellular space (Supplementary Fig. 2B) (below). It has been reported that the 30 amino acids at the C-terminus of Eno2 promote neuron survival, differentiation and axonal regeneration (21).

Eno1 is another member in the Enolase family located on the cell membrane, and it can interact with plasminogen through its exposed C-terminal lysine residue (22-23). Thus, we also used ProtScale-ExpASy software to analyze the hydrophobicity of Eno1, as you suggested. Similar to Eno2 (Supplementary Fig. 2B), the C-terminal sequence of Eno1 (Supplementary Fig. 2A) is also a hydrophilic structure (below). This evidence strengthens our hypothesis that Eno2, which is located on the cell membrane, may interact with secreted proteins, such as Pgk1, through its C-terminal hydrophilic structure.

Supplementary Fig. 1

Supplementary Fig. 2B

Finally, as you suggested, we obtained the X-ray structures of human Pgk1 (PDB ID: 2ZGV) and Eno2 (PDB ID: 5TD9) from the Protein Data Bank and proposed a structural model (below) that displays the interaction between ePgk1 and Eno2, using ZDOCK software (Pierce et al. 2014). Since Rider et al. (1974) and Suzuki et al. (1980) proposed that Eno2 is presented as a dimer, the topographical model shown below was based on this hypothesis, illustrating that PgK1 (blue) through PgK1 325-417 (orange) interacted with Eno2 404-431 (red) of Eno2 (green)

However, based on direct observation through immunoelectron microscopy (IEM) performed in this study, as shown in Figure 2, we randomly selected 9 signals from three signal-positive neural cells. We found that colocalization between Pgk1-Flag (18-nm gold particles) and Eno2 (12-nm gold particles) exhibited two patterns: (1) one large particle associated with one small particle (88.8 %, n=9) and (2) one large particle associated with two small particles (11.2 %, n=9). These results suggest that one molecule of ePgk1 ligand could interact with either one monomer (major type) or one dimer (minor type) of receptor Eno2. Thus, unlike the conventional concept of Eno2 presented as a dimer, direct evidence by IEM in this study demonstrated that Eno2 could also present as a monomer on the motor neuron cell membrane. Therefore, since the faithful topographical structure of the interaction between ePgk1 and Eno2 is still controversial at the present time, we prefer to pull the old Figure 8 in the revised manuscript just because this figure might be too simple to present the interaction between ePgk1 and Eno2.

References

- Pierce, B. G., et al. ZDOCK server: interactive docking prediction of protein-protein complexes and symmetric multimers. *Bioinformatics*, 30, 1771-1773. (2014).
- Rider, C.C. & Taylor, C.B. Enolase isoenzymes in rat tissues. Electrophoretic, chromatographic, immunological and kinetic properties. *Biochim. Biophys. Acta.* 365, 285-300. (1974).
- Suzuki, F., Umeda, Y. & Kato, K. Rat brain enolase isozymes. Purification of three forms of enolase. *J. Biochem.* 87, 1587-1594. (1980).

(2) NSC34 cells are useful for biochemical analysis but are not necessarily the best system to study axonal outgrowth. Primary neuronal cultures would be recommended. I appreciate this might require setting up assays that are currently unavailable in the Team's laboratories. Zebrafish embryos are used here as an *in vivo* model to study the effects of ePgk1/Eno2 on neurite extension. However, I am not convinced that the reported phenotypes (ectopic axon branching) are an indication of "increased neurite outgrowth of motor neurons". They seem more in line with a generic effect on cytoskeletal remodeling and possibly a perturbation of normal axon growth. In the absence of a clear readout for axon elongation, I would recommend the Authors are cautious with the interpretation of the phenotypes.

Author's response to (2):

Indeed, testing primary neuronal cultures is not currently available in our laboratories, as you predicted. However, in this study, we applied motor neurons derived from Neuroblastoma x Spinal Cord (NSC) hybrid cell line NSC34, as reported by Cashman et al. (1992), to study neurite outgrowth of zebrafish embryos derived from transgenic line Tg(mnx:GFP), according to studies by Lin et al. (2019), Nango et al. (2017) and Jung et al. (2016).

Regarding the zebrafish *in vivo* model, we referred to the work of Kalil and Dent (2014) who demonstrated that cell membrane protrusion during axonal branching requires that accumulated actin filaments (F-actin) form patches. Membrane protrusion also requires actin nucleation, actin branching and actin elongation. When actin bundles extend outward forming filopodia, microtubules are cleaved by Spastin and Katanin to then enter the nascent branch, which continues to mature and extend axonal branching. In this process, Cofilin is important for the turnover of actin filaments (Please see attached Figure 2 below from Kalil and Dent (2014)). Based on this previously reported evidence, we believe that the axonal branching phenotype observed in zebrafish embryos results from cytoskeletal remodeling caused by additional F-actin formation resulting from the decrease of phosphorylated Cofilin (p-Cofilin), a growth cone

collapse marker. Since we observed that motor neurons increased axonal branching in ePgl1/Eno2-overexpressed embryos, it was convincing that ePgl1/Eno2 could reduce p-Cofilin level, leading to an increase in the formation of actin filaments and resultant cytoskeletal remodeling, which, in turn, would increase axonal branching. Therefore, we conclude that axonal outgrowth, as shown in this study, is the main physiological outcome of signaling.

(Kalil and Dent, 2014; Figure 2)

References

- Cashman, N.R. et al. Neuroblastoma x spinal cord (NSC) hybrid cell lines resemble developing motor neurons. *Dev. Dyn.* 194, 209-221. (1992).
- Jung, N. et al. Tonsil-derived mesenchymal stem cells differentiate into a Schwann cell phenotype and promote peripheral nerve regeneration. *Int. J. Mol. Sci.* 17, 1867. (2016).
- Kalil, K., & Dent, E. W. Branch management: mechanisms of axon branching in the developing vertebrate CNS. *Nat. Rev.-Neurosci.*, 15, 7–18. (2014).
- Lin, C.Y. et al. Extracellular Pgl1 enhances neurite outgrowth of motoneurons through Nogo66/NgR-independent targeting of NogoA. *eLife* 8, e49175. (2019).
- Nango, H. et al. Prostaglandin E2 facilitates neurite outgrowth in a motor neuron-like cell line. NSC-34. *J. Pharmacol. Sci.* 135, 64-71. (2017).

Further, taking your advice to use caution when interpreting phenotypes, we set up experiments to capture images with a much wider field, followed by analysis of neurite cells and neurite length using the *Neurite Outgrowth Analysis Application* Module of MetaMorph software. In the revised manuscript, we added a revised description in M&M, a new Figure 5 (please see below), all 15 original photos taken for each group for one trial (Source data for Figure 5A; see below), and a revised legend of Figure 5, as follows: “(A) Morphology of neurites derived from NSC34 neural cells was observed under microscopy. Five experimental groups were categorized: cells treated with pCS2 transfection, pCS2 transfection plus Pgk1 incubation (pCS2+Pgk1), pCS2-Eno1 transfection plus Pgk1 incubation (pCS2-Eno1+Pgk1), pCS2-Eno2 transfection plus Pgk1 incubation (pCS2-Eno2+Pgk1) and pCS2-Eno3 transfection plus Pgk1 incubation (pCS2-Eno3+Pgk1). Scale bar, 100 μ m. (B) *Neurite Outgrowth Analysis Application* Module of MetaMorph software was used to count neurite-bearing cells and measure neurite lengths. Cell body was marked in white, while neurite was marked in red. (C-D) Statistical analysis: (C) percentage of neurite-bearing cells among examined cells and (D) average neurite length of each group, as indicated. Experiments were performed independently at three trials. Final data were presented as averaged from 3 trials.” (Please see Results section from line 1003 of page 45 to line 1014 of page 46)

M&M was revised, as follows: “Neurites extending from each NSC34 cell were captured under phase contrast microscopy (ZEISS Axio Observer Z1, Göttingen, Germany) at 10x magnification. Five experimental groups were established, as follows: cells treated with pCS2 transfection, pCS2 transfection plus Pgk1 incubation, pCS2-Eno1 transfection plus Pgk1 incubation, pCS2-Eno2 transfection plus Pgk1 incubation and pCS2-Eno3 transfection plus Pgk1 incubation. For each trial, 15 pictures were randomly taken for each experimental group, followed by analysis of neurite cells and neurite length using the *Neurite Outgrowth Analysis Application* Module of MetaMorph software. Cells with axons longer than 30 μ m were defined as neurite-bearing cells. The percentage of neurite-bearing cells among examined cells and the average length of each neurite were calculated. Datum from each trial was averaged from 15 photos. Final data were averaged from three independent experiments and expressed as mean \pm SD.” (Please see Results section from line 717 of page 32 to line 728 of page 32)

Fig. 5

Source data for Fig. 5A

(3) I was not sure how to interpret the effect of Eno2 blocking antibody (Fig. 7) and siRNA (Suppl. Fig. 4) in the absence of ePkg1 ligand. Can the Author provide an explanation?

Author's response to (3):

In response to your comment, we have revised the Results section subtitled “The signaling pathway involved in reducing p-Cofilin through the interaction between ePkg1 and membrane protein Eno2 in NSC34 cells” (line 261). The fourth paragraph (line 293) reads as follows: “We then collected the total proteins of NSC34 cells extracted from three sources, including (a) cells incubated with normal mouse IgG, which served as a control group; (b) cells incubated with Pkg1 and IgG; and (c) cells incubated with Pkg1 plus Eno2 antibody. Compared with control cells incubated with normal mouse IgG normalized as 1, the protein levels of p-P38-T180, p-Limk1-S323 and p-Cofilin were all significantly reduced in cells incubated with Pkg1 and IgG by 61, 30 and 43%, respectively (Fig. 7C-D). However, the levels of total proteins of P38, p-Limk1-T508, Limk1 and Cofilin remained relatively unchanged in the Pkg1 plus IgG cells (Fig. 7C-D), suggesting that the effect of ePkg1 on reducing p-Cofilin was functional in the presence of general antibody. Nevertheless, compared to the IgG control group, the levels of the above phosphorylated and total proteins remained nearly unchanged in cells incubated with Pkg1 plus Eno2 antibody, except that the levels of p-P38-T180 and p-Cofilin were reduced by 11 and 10%, respectively (Fig. 7C-D). Next, we extracted total proteins from (d) NSC34 cells incubated with Eno2 antibody alone, and the results showed that the protein levels of p-P38-T180, p-Limk1-S323 and p-Cofilin were all significantly increased (Fig. 7C-D) by 113, 77 and 62%, respectively, compared to those of control IgG-incubated cells. Collectively, these results showed that the reduction of p-P38/p-Limk1-S323/p-Cofilin axis by ePkg1 was mostly abolished in the presence of Eno2 antibody, further indicating that receptor Eno2 on neural membrane is specifically bound by ligand ePkg1 to trigger the signaling pathway that causes neurite outgrowth.

To further confirm this hypothesis, we employed *Eno2*-siRNA to specifically knock down endogenous Eno2, compared with cells transfected with control siRNA, the protein levels of p-P38-T180, p-Limk1-S323 and p-Cofilin were all significantly reduced by 58, 34 and 45%, respectively, while the total protein levels of P38, Limk1, and Cofilin, as well as p-Limk1-T508, remained nearly unchanged in cells incubated with Pkg1 and control siRNA (Supplementary Fig. 6A-B), suggesting that the reduction of p-P38/p-Limk1-S323/p-Cofilin axis by ePkg1 was functional in the presence of control siRNA. Moreover, we found that the levels of the above phosphorylated could be restored and total proteins remained unchanged in cells incubated with Pkg1 plus *Eno2*-siRNA except that p-Cofilin was reduced by 7% and p-P38 was increased by 14%

(Supplementary Fig. 6A-B), suggesting that the reduction of p-Cofilin by ePgk1 was nearly abolished by the presence of *Eno2*-siRNA. These results were consistent with those obtained from cells treated with *Eno2*-antibody, as described above. Meanwhile, when we extracted total proteins from NSC34 cells incubated with *Eno2*-siRNA alone, we found that the protein levels of p-P38-T180, p-Limk1-S323 and p-Cofilin were all significantly increased by 84, 59 and 54%, respectively, compared to those of control siRNA-transfected cells.” (Please see the Results section from line 309 of page 14 to line 344 of page 16)

We added the following new description to the Discussion: “To determine the effect of ePgk1 on neurite outgrowth in cells lacking the *Eno2* receptor, we performed knockdown experiments with both *Eno2*-antibody and *Eno2*-siRNA. Compared to the control IgG group normalized as 1 in the first experiment, we noticed that the levels of p-Cofilin, p-Limk1-S323 and p-P38 were reduced by 10, 1 and 11%, respectively, in cells incubated with Pgk1 plus *Eno2*-antibody (Fig. 7C-D). These results indicate that p-Cofilin and p-P38 were slightly reduced in cells in spite of blocking *Eno2* receptor by *Eno2*-antibody. Meanwhile, compared with control siRNA-transfected cells in the parallel siRNA-knockdown experiment, the reduction of p-P38-T180, p-Limk1-S323 and p-P38, as mediated by ePgk1, was almost abolished in cells incubated with Pgk1 plus *Eno2*-siRNA transfection (Supplementary Fig. 6A-B). Yet, we also noticed that p-Cofilin could only be reduced by 7%. Based on this evidence from both experiments, it can be concluded that the reduction of p-Cofilin, as mediated by ePgk1, is nearly, but not completely, abolished in the absence of *Eno2*, further supporting our hypothesis that ePgk1 ligand triggers the signaling pathway that causes neurite outgrowth by interacting with receptor *Eno2* on the neural membrane. Consequently, the interaction between ePgk1 and *Eno2* plays an important role in reducing p-Cofilin, even though it suffers only a slight reduction when *Eno2* receptor is blocked or knocked down. Based on these results, it is plausible that the dosage of either *Eno2*-siRNA or *Eno2*-antibody employed in this study could not completely eliminate the presence of endogenous *Eno2* in the treated cells. In fact, Western blot analysis demonstrated that *Eno2* extracted from the *Eno2*-siRNA-transfected NSC34 cells was reduced by only 80% (Supplementary Fig. 3A), notwithstanding that *Eno2*-siRNA is an effective blocker of *eno2* expression. This, in turn, increases the speculation that ePgk1 interacts with *Eno2* receptor remaining, even after blockage or knockdown, and that this remainder is sufficient for the reduction in p-Cofilin. It is also possible that ePgk1 affects the p-P38-T180/p-Limk1-S323/p-Cofilin pathway in a manner that departs from its major interaction with intracellular *Eno2*. In this scenario, additive ePgk1 could bind some minor, or alternative, as-yet unidentified receptor, or it could be transported intracellularly through endocytosis, to reduce p-Cofilin, in turn slightly favoring axonal

branching. Results from the branched neurite experiment might support this hypothesis since the percentage of branched neurites that occurred in embryos injected with *eno2*-MO plus Pgk1 was slightly higher than that of embryos injected with *eno2*-MO alone, but lower than that of embryos injected with ePgk1 alone (Supplementary Fig. 4B-D). However, only further study will confirm the above hypotheses.

Finally, we extracted total proteins from NSC34 cells, either incubated with Eno2-antibody or transfected with Eno2-siRNA, and analyzed the protein levels involved in the p-P38/p-Limk1/p-Cofilin pathway. Compared with control IgG-cells, we found that the protein level of the p-P38/p-Limk1-S323/p-Cofilin axis was increased to 113,77 and 62%, respectively, in Eno2-antibody-treated cells (Fig.7C-D). Similarly, the protein levels of p-P38/p-Limk1-S323/p-Cofilin were all significantly increased to 84,59 and 54%, respectively, in cells incubated with *Eno2*-siRNA alone (Supplementary Fig. 6A-B). Based on these results, it could be speculated that intracellular *Eno2 per se* may repress the p-P38-T180/p-Limk1-S323/p-Cofilin pathway through some unknown regulatory mechanism apart from its interaction with ePgk1. Therefore, during the *Eno2*-knockdown experiment, it is plausible that the remaining unsilenced *Eno2* might slightly attenuate p-Cofilin in a manner independent of its interaction with ePgk1, giving still a third explanation for the effect of ePgk1 on neurite outgrowth in cells lacking the *Eno2* receptor.” (Please see Discussion section from line 459 of page 21 to line 503 of page 23)

(4) Several points in the manuscript require attention:

(4A) - lines 269-276: the effect on p-Cofilin in NSC34 cells has already been presented for Suppl. Fig. 1. The two paragraphs could be combined to avoid redundancy.

Author’s response to (4A):

You may have misread the text. To determine which member of the Enolase family interacts with ePgk1 to reduce p-Cofilin in NSC34 cells, we first performed an *Eno1*- and *Eno2*-siRNA inhibition experiment. Data shown in Suppl. Fig. 1 showed a strong involvement between *Eno2* and ePgk1, which encouraged us to explore the possible synergism between ePgk1 and *Eno2*, as shown in Figure 7A. Suppl. Fig. 1 described a different issue from Figure 7A, so we presented the two issues separately.

(4B) - line 279: “decreased in the culture medium”. Is this a typo for “total protein lysate”?

Author’s response to (4B):

We apologize for making this mistake. We corrected it in the revised text, as follows: “We found that (a) the protein levels of p-P38-T180 and p-Limk1-S323 were decreased in total protein lysate of NSC34 cells, either incubated with ePgk1 or

transfected with Eno2; (b) the protein levels of p-P38-T180 and p-Limk1-S323 were synergistically decreased in the lysate of NSC34 cells treated with the combination of Pgk1 incubation and Eno2 mRNA transfection; and (c) the protein level of p-Limk1-T508 remained unchanged in the lysate of NSC34 cells treated with the combination of Pgk1 incubation and Eno2 mRNA transfection. (Please see the Results section from line 294 of page 13 to line 301 of page 14)

(4C) - lines 305-312 describe the efficiency of siRNA already presented for Suppl. Fig. 1. Could be removed to avoid redundancy.

Author's response to (4C):

Thank you. We removed it.

(4D) - line 314: one of the conditions should be si-eno2 only (without ePgk1)

Author's response to (4D):

Thank you for pointing out this mistake. We corrected it.

(4E) - I recommend shortening and streamlining the Discussion to avoid diluting the interpretation of the findings. What is the relevance of the section on Enolase family (line 347).

Author's response to (4E):

In response to your suggestion, we deleted the section on Enolase family. However, our textual response to 3 above was added.

Other points:

(1)- Suppl Fig 1: mark specific bands on WB (A,B)

Author's response to (1):

Thank you. We marked specific bands on Suppl Fig. 1 (A, B) (please see below) and added this description to the revised Suppl Fig. 1 legend as follows: "Location of the Eno1 band on Panel A was marked with an empty arrowhead, while the Eno2 band on Panel B was marked with a solid arrow." (Please see legends for Supplementary Fig. 3 from line 1116 of page 54 to line 1118 of page 54).

A

B

(2)- For how long were embryos incubated with recombinant ePgk1? I could not find in the Methods. It should be added to the legend.

Author's response to (2):

In response to your question, we added more detailed description in Materials and methods, as follows: “For incubation of recombinant zPgk1 in zebrafish embryos, we first purified the recombinant fusion protein zPgk1-Flag produced by HEK293T cells. The chorion of embryos at 6 hpf was partially removed by sharp forceps (Dumont No. 5), if necessary, followed by their immersion in embryo media containing zPgk1-Flag with a concentration of 300 ng/ml and incubation for 24 hr at 26~28°C. Then, embryos at 30 hpf were fixed with 4% PFA overnight, and their neurite outgrowth of motor neurons was observed under fluorescence microscopy.” (Please see Materials and methods section from line 701 of page 31 to line 707 of page 31).

We also added some statements in the legend, as follows: “(C) After embryos were incubated, either with (+) or without (-) zebrafish Pgk1 for 24 hr, they were injected with or without three isoforms of *eno*-mRNAs. The percentage of embryos displaying branched motor neuron phenotype among the total number (n) of examined embryos in each group was calculated. (Please see legends for Fig. 6 from line 1032 of page 47 to line 1035 of page 47).

(3)- What are the levels of Eno2 protein overexpression achieved after mRNA injection in embryos (Figure 6)?

Author's response to (3):

To address your question, we performed a new Western blot to analyze the level of Eno2 protein after injection of *eno2*-mRNA in embryos. Compared to the protein level

of Eno2 of WT zebrafish embryos normalized as 1, results showed that the protein level of Eno2 was significantly increased up to 384% in the *eno2-mRNA*-injected zebrafish embryos (see attached figure below).

(4)- Suppl. Figure 2 examines motor neuron phenotypes in *eno2*-MO embryos. The levels of p-cofilin could be measured as in Figure 6.

Author's response to (4):

In accordance with your suggestion, our analysis of phosphorylated Cofilin at S3 (p-Cofilin) and total Cofilin is expressed in the legend given in new Suppl. Figure 4, as follows: “(A) Control-MO and *eno2*-MO were individually injected into one-celled embryos. Antibody against Eno2 was used to perform Immunoblotting (IB) analysis with the aim of confirming the effective and specific knockdown by *eno2*-MO. The relative levels of Eno2, phosphorylated Cofilin at S3 (p-Cofilin) and total Cofilin were quantified based on the values compared to those of control MO set as 1 and shown on each lane below. The α -tubulin served as an internal control.” (Please see legends for Supplementary Fig. 4 from line 1145 of page 56 to line 1150 of page 57)., and in revised text, as follows: “In comparison to control MO-injected zebrafish embryos, was normalized as 1, Western blot analysis demonstrated that the protein level of *eno2* was reduced to 20% and the protein level of p-Cofilin was significantly increased to 254% in the *eno2*-MO-injected zebrafish embryos (Supplementary Fig. 4A), these suggesting that *eno2*-MO is specific and effective in knocking down the expression of *eno2* further to resulting increase of p-Cofilin.” (Please see the Results section from line 259 of page 12 to line 264 of page 12).

Referee #3

Overexpression of NogoA in muscle cells is known to positively correlate with the loss of motor endplates in diseases such as Amyotrophic Lateral Sclerosis. NogoA-overexpressing skeletal muscle cells have the capacity to reduce secretion of phosphoglycerate kinase 1 (Pgk1), thus increasing p-Cofilin, a growth cone collapse marker, by triggering p-P38-T180/p-Limk1-S323 signals in motoneurons and attenuating the neurite outgrowth of motoneurons (NOMN). Thus, the authors investigated whether NOMN could be controlled by extracellular Pgk1 (ePgk1) plus an interacting neural receptor which is not currently known. The authors found that Enolase-2 (Eno2) can exhibit strong affinity with recombinant Pgk1-Flag, and the 325th-417th amino acids of ePgk1 may interact with the 404th-431st amino acids of Eno2. This is convincing. However, the role of Eno1 is still not clear although the authors showed some data suggesting neither Eno1 nor Eno3 enhanced NOMN in NSC34 cells and zebrafish embryos. There may be a minor role of Eno1 since Eno1 has the dominant function on the surface. The function of Eno3 is somewhat supported by the data. However, the role of Eno2 could be dominant as it is strongly supported by the data.

(1A) Figure 6 and 7, Western blot data. Some of the blots are not clear. Figure 6 Cofilin data are not convincing.

Author's response to (1A):

In accordance with your suggestion, we improved the quality of Figures 6 and 7 (please see revised Figures 6 and 7 below).

Fig. 6

D

E

Fig. 7

(1B) In Figure 7, bands with similar molecular weight are presented with a single loading control. This creates confusion.

Author's response to (1B):

This did not concern us. In fact, when we performed the Western blot experiment, the prepared samples were dispensed for multiple groups of loading to carry out SDS-PAGE simultaneously. After electrophoresis, we separately performed Western blot analysis for each group of loading using one specific antiserum. For example, one group of loading samples was used to detect Limk1-S323 only, while the next group of loading samples was used to detect Limk1. Since immunoblot staining was performed separately, bands with similar molecular weight are not overlapped and,

thus, cannot be confused. For example, the molecular weight of Limk1-S323, Limk1-T508 and Limk1 are all around 72 kDa, but each band was clearly presented on each group of loading sample. (Please see Source data for Fig. 7 attached below)

Source data for Fig. 7

Figure 7A

Figure 7C

(2) The authors utilized immunoprecipitation and LC-MS/MS analyses to screen putative membrane receptors of ePgk1. Studies here with the screening and interaction of proteins in cell lines and zebrafish embryos convincingly showed that Eno2 had the strongest affinity with Pgk1, and may regulate the P38/Limk1/Cofilin signaling axis, improving the neurite outgrowth of motor neurons. Data presented here support association of ePgk1 with Eno2 on the cell membrane of neural cells, and the 404th-431st amino acid residues of Eno2 may comprise the key domain that interacts with ePgk1. Data also support the finding that the 325-417 domain of ePgk1 could be the key domain interacting with Eno2. However, the role of Eno1 should be revisited as this reviewer thinks that Eno1 may have a minor role in the interaction with ePgk1.

Author's response to (2):

In response to your question, we first designed a new experiment to determine whether ePgk1 could interact with Eno1. Based on the results from the cell surface crosslinking-immunoprecipitation experiment (see figure below), ePgk1 can interact with Eno2, but not Eno1.

Second, based on *in vitro* (see Fig. 5 shown below) and *in vivo* systems (see Fig. 6 shown below), we demonstrated that ePgk1 affected motor neuron development through its interaction with Eno2, neither Eno1 nor Eno3.

Fig. 5

Fig. 6

Third, compared with cells incubated with ePgc1, our siRNA-knockdown experiment (see figure below; Supplementary Fig. 3C-D) demonstrated that the protein level of p-Cofilin remained unchanged in cells transfected with *Eno1*-siRNA and incubated with Pgc1, while the protein level of p-Cofilin was increased in cells transfected with *Eno2*-siRNA and incubated with Pgc1, suggesting that the reduction of p-Cofilin induced by ePgc1 resulted from specific interaction between ePgc1 and *Eno2*, not *Eno1*.

Supplementary Fig. 3

Taken together, the evidence from three experiments described above leads to the conclusion that *Eno2* is located on the cell membrane of motor neurons as a receptor and that it interacts with ligand ePgk1 to promote the neurite outgrowth of motor neurons through the signaling pathway proposed in the original version of this article.

REVIEWERS' COMMENTS:

Reviewer #1 (Remarks to the Author):

The authors completed all the critiques raised by this reviewer.
No further issues are found with the revised manuscript.

Reviewer #2 (Remarks to the Author):

I appreciate the Authors' efforts to address my concerns by performing additional experiments and revising the manuscript. I understand that some of the suggested experiments required setting up assays that are not currently available in the Authors' lab (ie, primary neuronal cultures) and I agree that the results obtained with NSC34 cells are acceptable - although not ideal in my opinion. The paper provides interesting insights into the non-canonical extracellular function of Pgk1 and I expect it to be of general interest. Therefore I recommend publication without further amendment.